# DRIBO: Robust Deep Reinforcement Learning via Multi-View Information Bottleneck

## Abstract

Deep reinforcement learning (DRL) agents are often sensitive to visual changes that were unseen in their training environments. To address this problem, we leverage the sequential nature of RL to learn robust representations that encode only task-relevant information from observations based on the unsupervised multi-view setting. Specifically, we introduce a novel contrastive version of Multi-View Information Bottleneck (MIB) objective for temporal data. We train RL agents from pixels with this auxiliary objective to learn robust representations that can compress away task-irrelevant information and are predictive of task-relevant dynamics. This approach enables us to train high-performance policies that are robust to visual distractions and can generalize well to unseen environments. We demonstrate that our approach can achieve SOTA performance on diverse visual control tasks on the DeepMind Control Suite when the background is replaced with natural videos. In addition, we show that our approach outperforms well-established baselines for generalization to unseen environments on the Procgen benchmark.

## 1 Introduction

Deep reinforcement learning (DRL) methods have been shown to be successful in learning high-quality controllers directly from raw images in an end-to-end fashion (Mnih et al., 2015; Levine et al., 2016; Bojarski et al., 2016). However, it has been observed that DRL agents perform poorly in environments different from those where the agents were trained on, even when these environments contain semantically equivalent information relevant to the control task (Farebrother et al., 2018; Cobbe et al., 2019; Zhang et al., 2018b;a; Yu et al., 2019). By contrast, humans routinely adapt to new, unseen environments. For example, while visual scenes can be drastically different when driving in different cities, human drivers can quickly adjust to driving in a new city which they have never visited. We argue that this ability to adapt stems from the fact that driving skills are invariant to many visual details that are actually not relevant to driving. Conversely, DRL agents without this ability are hindered from understanding the temporal structure of *task-relevant* dynamics without being distracted by *task-irrelevant* visual details (Jonschkowski & Brock, 2015; Zhang et al., 2021; Agarwal et al., 2021; Lee et al., 2020b).

Viewing from a representation learning perspective, a desired representation for RL should facilitate the prediction of future states (beyond expected rewards) on potential actions and discard excessive, task-irrelevant information from visual observations. An RL agent that learns from such representations has the advantage of learning an optimal policy more easily upon the prediction and being more *robust to visual changes*. In addition, the resulting policy is more likely to *generalize to unseen environments* if the task-relevant information in the new environment remains similar to that in the training environments. Prior works (Hafner et al., 2019; Lee et al., 2020a) that encode images into a low-dimensional latent space for RL typically rely on a reconstruction loss to learn representations that are sufficient to reconstruct the input images and predict ahead in the latent space While these approaches can learn representations that retain information in the visual observations, they do nothing to discard the irrelevant information.

We tackle this problem by considering state representations for RL that are robust under the multi-view setting (Li et al., 2018; Federici et al., 2020; Fischer, 2020), where each view is assumed to provide the same amount of *task-relevant* information while all the information not shared by them is deemed *task-irrelevant*. Data augmentation can be easily leveraged to generate such multi-view observations

Figure 1: Robust **D**eep **R**einforcement Learning via Multi-View **I**nfomration **BO**ttleneck (DRIBO) incorporates the inherent temporal structure of RL and unsupervised multi-view settings into robust representation learning in RL. We consider sequential multi-view observations, $o_{1:T}^{(1)}$ and $o_{1:T}^{(2)}$, of original observations $o_{1:T}$ sharing the same *task-relevant* information while any information not shared by them are *task-irrelevant*. For example, natural video backgrounds of sequential observations are task-irrelevant and can be drastically different between training and testing environments. DRIBO uses a multi-view information bottleneck loss to ensure that $s_{1:T}^{(1)}$ and $s_{1:T}^{(2)}$, the representations of multi-view observations, shares maximal task-relevant information while reducing the task-irrelevant information. DRIBO trains the RL policy and (or) value function on top of the encoder.

without requiring additional new data. Data augmentation in RL has delivered promising results for visual control tasks (Laskin et al., 2020b; Lange et al., 2012; Laskin et al., 2020a). However, these methods rarely exploit the sequential aspect of RL which requires an ideal representation to be predictive of future states given actions. In fact, the sequential nature of RL provides an additional temporal dimension for identifying *task-irrelevant* information when it is independent of actions. Instead of learning representations from each visual observation (Laskin et al., 2020a), we propose to learn a predictive model that captures the temporal evolution of representations from a sequence of observations and actions. Concretely, ***we introduce a new multi-view information bottleneck (MIB) objective that maximizes the mutual information between sequences of observations and representations while reducing the task-irrelevant information identified from the multi-view observations.*** We incorporate this MIB objective into RL by using it as an auxiliary learning objective. We illustrate our approach in Figure 1. Our contributions are summarized below.

- We propose DRIBO, a novel technique that learns robust representations in RL by identifying and discarding *task-irrelevant* information in the representations based on MIB.
- We leverage the sequential nature of RL to learn representations better suited for RL with a non-reconstruction-based, DRIBO loss that maximizes the mutual information between sequences of observations and representations while disregarding *task-irrelevant* information.
- Empirically, we show that our approach can (i) lead to better robustness against task-irrelevant distractors on the DeepMind Control Suite and (ii) significantly improve generalization on the Procgen benchmarks compared to current state-of-the-arts.

## 2 RELATED WORK

**Reconstruction-based Representation Learning.** Early works first trained autoencoders to learn representations to reconstruct raw observations. Then, the RL agent was trained from the learned representations (Lange & Riedmiller, 2010; Lange et al., 2012). However, there is no guarantee that the agent will capture useful information for control. To address this problem, learning encoder and dynamics jointly has been proved effective in learning task-oriented and predictive representations (Wahlström et al., 2015; Watter et al., 2015). More recently, Hafner et al. (2019; 2020; 2021) and Lee et al. (2020a) learn a latent dynamics model and train RL agents with predictive latent representations. However, these approaches suffer from the problem of embedding all details into representations even when they are task-irrelevant. The reason is that improving reconstruction quality from representations to visual observations forces the representations to retain more details. Despite success on many benchmarks, task-irrelevant visual changes can affect performance significantly (Zhang et al., 2018a). Experimentally, we show that our non-reconstructive approach, DRIBO, is substantially more robust against visual changes than prior works. We also compare DRIBO with another non-reconstructive method, DBC (Zhang et al., 2021), which uses bisimulation metrics to learn representations in RL that contain only task-relevant information.

**Contrastive Representations Learning.** Contrastive representation learning methods train an encoder that obeys similarity constraints in a dataset typically organized by similar and dissimilar

pairs. The similar examples are typically obtained from nearby image patches (Oord et al., 2018; Hénaff et al., 2020) or through data augmentation (Chen et al., 2020). A scoring function that lower-bounds mutual information is one of the typical objects to be maximized (Belghazi et al., 2018; Oord et al., 2018; Hjelm et al., 2019; Poole et al., 2019). A number of works have applied the above ideas to RL settings to extract predictive signals. EMI (Kim et al., 2019) applies a Jensen-Shannon divergence-based lower bound on mutual information across subsequent frames as an exploration bonus. DRIML (Mazoure et al., 2020) uses an auxiliary contrastive objective to maximize concordance between representations to increase predictive properties of the representations conditioned on actions. CURL (Laskin et al., 2020a) incorporates contrastive learning into RL algorithms to maximize similarity between augmented versions of the same observation. However, solely maximizing the lower-bound of mutual information retains all the information including those that are task-irrelevant (Federici et al., 2020; Fischer, 2020).

**Multi-View Information Bottleneck (MIB).** MVRL (Li et al., 2019) uses the multi-view setting to tackle partially observable Markov decision processes with more than one observation model. For classification tasks, Federici et al. (2020) uses MIB by maximizing the mutual information between the representations of the two views while at the same time eliminating the label-irrelevant information identified by multi-view observations. Fischer (2020) describes a variant of the Conditional Entropy Bottleneck (CEB) which is mathematically equivalent to MIB. However, MIB/CEB cannot be directly used in RL settings due to the sequential nature of decision making problems. PI-SAC (Lee et al., 2020b) uses a contrastive version of CEB to model Predictive Information (Bialek & Tishby, 1999) which is the mutual information between the past and the future to solve RL problems. However, this approach does not scale to long sequential data in RL and in practice only models short-term Predictive Information. Task-relevant information in RL is relevant because they influence not only current control decision and reward but also states and rewards well into the future. Our work, DRIBO, learns robust representations with a predictive model to maximize the mutual information between sequences of representations and observations, while eliminating task-irrelevant information based on the information bottleneck principle. Learning a predictive model also adopts richer learning signals than those provided by individual observation and reward alone. Philosophically and technically, our approach is different from PI-SAC which does not quantify task-irrelevant information from multi-view observations and cannot capture long-term dependencies. Another line of work, IDAAC (Raileanu & Fergus, 2021), leverages an adversarial framework so that the learned representations yield features that are instance-independent and invariant to task-irrelevant changes.

## 3 PRELIMINARIES

We denote a Markov decision process (**MDP**) as $\mathcal{M}$, with state $s$, action $a$, and reward $r$. $S$ and $A$ stand for the corresponding random variables. We denote a policy on $\mathcal{M}$ as $\pi$. The agent's goal is to learn a policy $\pi$ that maximizes the cumulative rewards. We define $\mathcal{S} \subseteq \mathbb{R}^d$ as the state-representation space. The visual observations are $o \in \mathcal{O}$, where we denote multi-view observations from the viewpoint $i$ as $o^{(i)}$. $O$ stands for the random variable of the observation. We introduce a multi-view trajectory $\tau^{\mathcal{M}} = [s_1, o_1^{(i)}, a_1, \ldots, s_T, o_T^{(i)}, a_T]$ where $T$ is the length. Knowing that the trajectory density is defined over joint observations, states, and actions, we write:

$$p_\pi(\tau^{\mathcal{M}}) = \pi(a_T|s_T)\mathcal{P}_{\text{obs}}^{(i)}(o_T^{(i)}|s_T)\mathcal{P}(s_T|s_{T-1}, a_{T-1}) \cdots \pi(a_1|s_1)\mathcal{P}_{\text{obs}}^{(i)}(o_1^{(i)}|s_1)\mathcal{P}_0(s_1) \quad (1)$$

with $\mathcal{P}_0$ being the initial state distribution, $\mathcal{P}$ being the transition model and $\mathcal{P}_{\text{obs}}^{(i)}$ being the unknown observation model for view $i$. DRL agents learn from visual observations by treating consecutive observations as states to implicitly capture the predictive property. However, rich details in observations can easily distract the agent. An ideal representation should contain no task-irrelevant information and satisfy some underlying MDP which determines the distribution of the multi-view trajectory in Eq. (1). Thus, instead of mapping a single-step observation to a representation, we consider learning a predictive model that correlates sequential observations and representations.

Let $a_{1:T}^*$ be the *optimal* action sequence for some $o_{1:T}$ which is obtained by executing the action sequence $a_{1:T}$. We assume that $o_{1:T}$ contains enough information to obtain $a_{1:T}^*$ which maximizes the cumulative rewards. With this assumption, we define *task-relevant* information if it is necessary for deriving $a_{1:T}^*$. By contrast, *task-irrelevant* information does not contribute to the choice of $a_{1:T}^*$. We first consider *sufficient* representations that are discriminative enough to obtain $A^*$ at each

timestep. This property can be quantified by the amount of mutual information between $O_{1:T}$ and $A_{1:T}^*$ and mutual information between $S_{1:T}$ and $A_{1:T}^*$.

**Definition 1.** Representations $S_{1:T}$ of $O_{1:T}$ are *sufficient* for RL iff $\mathcal{I}(O_{1:T}; A_{1:T}^*)=\mathcal{I}(S_{1:T}; A_{1:T}^*)$.

RL agents that have access to a sufficient representation $S_t$ at timestep $t$ must be able to generate $A_t^*$ as if it has access to the original observations. This can be better understood by subdividing $\mathcal{I}(O_{1:T}; S_{1:T})$ into two components using the chain rule of mutual information:

$$\mathcal{I}(O_{1:T}; S_{1:T}) = \mathcal{I}(S_{1:T}; O_{1:T}|A_{1:T}^*) + \mathcal{I}(S_{1:T}; A_{1:T}^*) \tag{2}$$

Conditional mutual information $\mathcal{I}(S_{1:T}; O_{1:T}|A_{1:T}^*)$ quantifies the information in $S_{1:T}$ that is *task-irrelevant*. $\mathcal{I}(S_{1:T}; A_{1:T}^*)$ quantifies *task-relevant* information that is accessible from $S_{1:T}$. The last term is independent of the representation as long as $S_t$ is sufficient for $A_t^*$ (see Definition 1). Thus, a representation contains minimal task-irrelevant information whenever $\mathcal{I}(O_{1:T}; S_{1:T}|A_{1:T}^*)$ is minimized. Maximizing $\mathcal{I}(O_{1:T}; S_{1:T})$ learns a sufficient representation. With the information bottleneck principle (Tishby et al., 2000), we can construct an objective to maximize $\mathcal{I}(O_{1:T}; S_{1:T})$ while minimizing $\mathcal{I}(S_{1:T}; O_{1:T}|A_{1:T}^*)$ to compress away task-irrelevant information.

However, estimating the mutual information between long sequences is difficult due to the high dimensionality of the problem. In addition, the minimization of $\mathcal{I}(S_{1:T}; O_{1:T}|A_{1:T}^*)$ can only be done directly in supervised settings where $A_{1:T}^*$ are observed. One option is to use MIB which can compress away task-irrelevant information in the representations in unsupervised settings (Federici et al., 2020). The problem, however, is that MIB in its original form only considers a single observation and its representation and thus does not guarantee that the learned representations retain the important temporal structure of RL. In the next section, we describe how we extend MIB to RL settings.

# 4 DRIBO

DRIBO learns robust representations that are predictive of future representations while discarding task-irrelevant information. To learn such representations, we construct a new MIB objective that (i) reduces the problem of maximizing the mutual information between sequences of observations and representations to maximizing the mutual information between them at each timestep and (ii) quantifies the amount of task-irrelevant information in the representations in the multi-view setting.

## 4.1 MUTUAL INFORMATION MAXIMIZATION

To capture the temporal evolution of observations and representations given any action sequence, we consider maximizing the conditional mutual information $\mathcal{I}(S_{1:T}; O_{1:T}|A_{1:T})$ which is a lower bound of $\mathcal{I}(O_{1:T}; S_{1:T})$ (see Appendix A.1). The observations $O_{1:T}$ are generated sequentially in the environment by executing the actions $A_{1:T}$. The conditional mutual information not only estimates the sufficiency of the representations but also maintains the temporal structure of RL problems.

To tackle the challenges of estimating the mutual information between sequences, we first factorize the mutual information between two sequential data to the mutual information at each timestep.

**Theorem 1.** *Let $O_{1:T}$ be the sequential observations obtained by executing action sequence $A_{1:T}$. If $S_{1:T}$ is a sequence of sufficient representations for $O_{1:T}$, we have:*

$$\mathcal{I}(S_{1:T}; O_{1:T}|A_{1:T}) \geq \sum_{t=1}^{T} \mathcal{I}(S_t; O_t|S_{t-1}, A_{t-1}) \tag{3}$$

The proof is included in Appendix A.1. Theorem 1 shows that the sum of mutual information $\mathcal{I}(S_t; O_t|S_{t-1}, A_{t-1})$ over multiple timesteps is a lower bound of $\mathcal{I}(S_{1:T}; O_{1:T}|A_{1:T})$. $\mathcal{I}(S_t; O_t|S_{t-1}, A_{t-1})$ is defined with conditional probabilities, $p(\boldsymbol{s}_t, \boldsymbol{o}_t|\boldsymbol{s}_{t-1}, \boldsymbol{a}_{t-1})$, $p(\boldsymbol{s}_t|\boldsymbol{s}_{t-1}, \boldsymbol{a}_{t-1})$ and $p(\boldsymbol{o}_t|\boldsymbol{s}_{t-1}, \boldsymbol{a}_{t-1})$. This lower bound models the dynamics and temporal structure of RL since the first conditional probability is the composition of transition probability and observation model and the second conditional probability is the transition probability of the underlying MDP. Thus, the factorized mutual information explicitly retains the predictive property of representations. Even when the representations $S_{1:T}$ are not sufficient, maximizing the mutual information between $S_t$ and $O_t$ ($\mathcal{I}(S_t; O_t|S_{t-1}, A_{t-1})$) encourages encoding more detailed features from $O_t$ into $S_t$ and makes $S_t$ sufficient.

## 4.2 MULTI-VIEW SETTING

To learn sufficient representations with minimal task-irrelevant information, we consider a two-view setting to identify the task-irrelevant information without supervision. Consider $o_t^{(1)}$ and $o_t^{(2)}$ to be two visual images of the control scenario from two different viewpoints. Under the multi-view assumption, any representation $s_t$ containing all information accessible from both views and is predictive of future representations would contain sufficient task-relevant information. Furthermore, if $s_t$ captures only the details that are visible from both observations, it would eliminate the view-specific details and reduce the sensitivity of the representation to view-changes.

A sufficient representation in RL retains all the information that is shared by mutually redundant observations $O_t^{(1)}$ and $O_t^{(2)}$. We refer to Appendix A for the sufficiency condition of representations and mutually redundancy condition (Federici et al., 2020) between $O_t^{(1)}$ and $O_t^{(2)}$. Intuitively, with the mutual redundancy condition, any representation that contains all the information shared by both views is as task-relevant as the joint observation. By factorizing the mutual information between $S_t^{(1)}$ and $O_t^{(1)}$ as in Eq. (2), we can identify two components:

$$\mathcal{I}(S_t^{(1)}; O_t^{(1)} | S_{t-1}^{(1)}, A_{t-1}) = \mathcal{I}(S_t^{(1)}; O_t^{(1)} | S_{t-1}^{(1)}, A_{t-1}, O_t^{(2)}) + \mathcal{I}(O_t^{(2)}; S_t^{(1)} | S_{t-1}^{(1)}, A_{t-1}) \quad (4)$$

Here, $S_{t-1}^{(1)}$ is a representation of visual observation $O_{t-1}^{(1)}$. Since we assume mutual redundancy between the two views, the information shared between $O_t^{(1)}$ and $S_t^{(1)}$ conditioned on $O_t^{(2)}$ must be irrelevant to the task, which can be quantified as $\mathcal{I}(S_t^{(1)}; O_t^{(1)} | S_{t-1}^{(1)}, A_{t-1}, O_t^{(2)})$ (first term in Eq. (4)). Then, $\mathcal{I}(O_t^{(2)}; S_t^{(1)} | S_{t-1}^{(1)}, A_{t-1})$ has to be maximal if the representation is sufficient. A formal description of the above statement can be found in Appendix A.

The less the two views have in common, the less task-irrelevant information can be encoded into the representations without violating sufficiency, and consequently, the less sensitive the resulting representation is to task-irrelevant nuisances. In the extreme, $s_t^{(1)}$ is the underlying states of MDP if $o_t^{(1)}$ and $o_t^{(2)}$ share only task-relevant information. With Eq. (3) and 4, we have $\mathcal{L}_{\text{IB}}^{(1)}$ to maximize the mutual information between representations and observations and compress away task-irrelevant information $\mathcal{I}(O_t^{(2)}; S_t^{(1)} | S_{t-1}^{(1)}, A_{t-1})$ based on the information bottleneck principle. $\lambda_1$ is a Lagrange multiplier. This loss also retains the temporally evolving information of the underlying dynamics.

$$\mathcal{L}_{\text{IB}}^{(1)} = -\sum_t (\mathcal{I}(S_t^{(1)}; O_t^{(1)} | s_{t-1}^{(1)}, A_{t-1}, O_t^{(2)}) - \lambda_1 \mathcal{I}(O_t^{(2)}; S_t^{(1)} | S_{t-1}^{(1)}, A_{t-1})) \quad (5)$$

Symmetrically, we define a loss $\mathcal{L}_{\text{IB}}^{(2)}$ for representations and observations from view 2:

$$\mathcal{L}_{\text{IB}}^{(2)} = -\sum_t (\mathcal{I}(S_t^{(2)}; O_t^{(2)} | s_{t-1}^{(2)}, A_{t-1}, O_t^{(1)}) - \lambda_2 \mathcal{I}(O_t^{(1)}; S_t^{(2)} | S_{t-1}^{(2)}, A_{t-1})) \quad (6)$$

The above losses extend MIB to RL and minimizing it learns representations that are robust to task-irrelevant distractors and predictive of the future. The multi-view observations can be easily obtained with random data augmentation techniques so that each view is augmented differently.

## 4.3 DRIBO LOSS FUNCTION

By re-parameterizing the Lagrangian multipliers (details in Appendix B), the average of two loss functions $\mathcal{L}_{\text{IB}}^{(1)}$ and $\mathcal{L}_{\text{IB}}^{(2)}$ from two views at timestep $t$ can be upper bounded as follows:

$$\mathcal{L}_t(\theta; \beta) = -\mathcal{I}_\theta(S_t^{(1)}; S_t^{(2)} | S_{t-1}, A_{t-1}) + \quad (7)$$
$$\beta D_{\text{SKL}}(p_\theta(s_t^{(1)} | o_t^{(1)}, s_{t-1}^{(1)}, a_{t-1}) || p_\theta(s_t^{(2)} | o_t^{(2)}, s_{t-1}^{(2)}, a_{t-1}))$$

where $\theta$ denotes the parameters of an encoder $p_\theta(s_t^{(1)} | o_t^{(1)}, s_{t-1}, a_{t-1})$ (details in Section 4.4), $s_{t-1}$ is a sufficient representation, $D_{\text{SKL}}$ represents the symmetrized KL divergence obtained by averaging the expected values of $D_{\text{KL}}(p_\theta(s_t^{(1)} | o_t^{(1)}, s_{t-1}^{(1)}, a_{t-1}) || p_\theta(s_t^{(2)} | o_t^{(2)}, s_{t-1}^{(2)}, a_{t-1}))$, $D_{\text{KL}}(p_\theta(s_t^{(2)} | o_t^{(2)}, s_{t-1}^{(2)}, a_{t-1}) || p_\theta(s_t^{(1)} | o_t^{(1)}, s_{t-1}^{(1)}, a_{t-1}))$, and the coefficient $\beta$ represents the trade-off between sufficiency and sensitivity to task-irrelevant information. $\beta$ is a hyper-parameter.

---

**Algorithm 1** DRIBO Loss

---

1: **Input**: Batch $\mathcal{B}$ storing $N$ sequential observations and actions with length $T$ from replay buffer.
2: Apply random augmentation transformations on $\mathcal{B}$ to obtain multi-view batches $\mathcal{B}^{(1)}$ and $\mathcal{B}^{(2)}$.
3: **for** $i, (\boldsymbol{o}_{1:T}^{(1)}, \boldsymbol{o}_{1:T}^{(2)}, \boldsymbol{a}_{1:T})$ in enumerate $(\mathcal{B}^{(1)}, \mathcal{B}^{(2)})$ **do**
4:     **for** $t = 1$ to $T$ **do**
5:         $\boldsymbol{s}_t^{(1)} \sim p_\theta(\boldsymbol{s}_t^{(1)}|\boldsymbol{o}_t^{(1)}, \boldsymbol{s}_{t-1}^{(1)}, \boldsymbol{a}_{t-1}), \boldsymbol{s}_t^{(2)} \sim p_\theta(\boldsymbol{s}_t^{(2)}|\boldsymbol{o}_t^{(2)}, \boldsymbol{s}_{t-1}^{(2)}, \boldsymbol{a}_{t-1})$
6:         $(\boldsymbol{s}^{(1),t+T(i-1)}, \boldsymbol{s}^{(2),t+T(i-1)}) \leftarrow (\boldsymbol{s}_t^{(1)}, \boldsymbol{s}_t^{(2)})$
7:     **end for**
8:     $\mathcal{L}_{\text{SKL}}^i = \frac{1}{T} \sum_{t=1}^T D_{\text{DKL}}(p_\theta(\boldsymbol{s}_t^{(1)}|\boldsymbol{o}_t^{(1)}, \boldsymbol{s}_{t-1}^{(1)}, \boldsymbol{a}_{t-1})||p_\theta(\boldsymbol{s}_t^{(2)}|\boldsymbol{o}_t^{(2)}, \boldsymbol{s}_{t-1}^{(2)}, \boldsymbol{a}_{t-1}))$
9: **end for**
10: **return** $-\hat{I}_\psi(\{(\boldsymbol{s}^{(1),i}, \boldsymbol{s}^{(2),i})\}_{i=1}^{T*N}) + \frac{\beta}{N} \sum_{i=1}^N \mathcal{L}_{\text{SKL}}^i$

---

To generalize the above loss to sequential data in RL, we apply Theorem 1 to obtain the DRIBO loss: $\mathcal{L}_{\text{DRIBO}} = \frac{1}{T} \sum_{t=1}^T \mathcal{L}_t(\theta; \beta)$. We summarize the batch-based computation of the loss function in Algorithm 1. We sample $\boldsymbol{s}_t^{(1)}$ and $\boldsymbol{s}_t^{(2)}$ from $p_\theta(\boldsymbol{s}_t^{(1)}|\boldsymbol{o}_t^{(1)}, \boldsymbol{s}_{t-1}^{(1)}, \boldsymbol{a}_{t-1})$ and $p_\theta(\boldsymbol{s}_t^{(2)}|\boldsymbol{o}_t^{(2)}, \boldsymbol{s}_{t-1}^{(2)}, \boldsymbol{a}_{t-1})$ respectively. Though the first term in Eq. (7) is conditioned on $\boldsymbol{s}_{t-1}$, we prove that the sampling process does not affect its effectiveness based on the multi-view assumption in Appendix B. The symmetrized KL divergence term can be computed from the probability density of $S_t^{(1)}$ and $S_t^{(2)}$ using the encoder. The mutual information between the two representations $\mathcal{I}_\theta(S_t^{(1)}; S_t^{(2)}|S_{t-1}, A_{t-1})$ can be maximized by using any sample-based differentiable mutual information lower bound $\hat{I}_\psi(\boldsymbol{s}_t^{(1)}, \boldsymbol{s}_t^{(2)})$, where $\psi$ represents the learnable parameters. We use InfoNCE (Oord et al., 2018) to estimate mutual information since the multi-view setting provides a large number of negative examples. The positive pairs are the representations $(\boldsymbol{s}_t^{(1)}, \boldsymbol{s}_t^{(2)})$ of the multi-view observations generated from the same observation. The remaining pairs of representations within the same batch are used as negative pairs. The full derivation of the DRIBO loss function can be found in Appendix B.

### 4.4 ENCODER ARCHITECTURE AND INCORPORATING DRIBO IN RL

The encoder $p_\theta(\boldsymbol{s}_t|\boldsymbol{o}_t, \boldsymbol{s}_{t-1}, \boldsymbol{a}_{t-1})$ approximates the posterior representation given the current observation, and representation and action from the previous timestep. The posteriors can also be viewed as a reparameterization of $p_\theta(\boldsymbol{s}_{1:T}|\boldsymbol{o}_{1:T}, \boldsymbol{a}_{1:T}) = \prod_t p_\theta(\boldsymbol{s}_t|\boldsymbol{o}_t, \boldsymbol{s}_{t-1}, \boldsymbol{a}_{t-1})$, which reflects the inherent temporal structure of RL. We implement the encoder as a recurrent space model (RSSM (Hafner et al., 2019)) which leverages recurrent neural networks to perform accurate long-term predictions. More details can be found in Appendix D.1. Training an RSSM encoder with DRIBO enables the representations to be predictive of future states.

We simultaneously train our representation learning models and the RL agent by adding $\mathcal{L}_{\text{DRIBO}}$ (Algorithm 1) as an auxiliary objective during training. The multi-view observations can be easily obtained using the same experience replay of RL agents through data augmentation. We demonstrate the effectiveness of DRIBO by building the agents on top of SAC (Haarnoja et al., 2018), an off-policy RL algorithm, and PPO (Schulman et al., 2017), an on-policy RL algorithm, in Section 5.1 and Section 5.3 respectively. More details can be found in Appendix D.

## 5 EXPERIMENTS

We experimentally evaluate DRIBO on a variety of visual control tasks. We designed the experiments to compare DRIBO to the current best methods in the literature on: (i) the effectiveness of solving visual control tasks, (ii) their robustness against task-irrelevant distractors, and (iii) the ability to generalize to unseen environments. For effectiveness and robustness, we demonstrate DRIBO's performance on DeepMind Control Suite (DMC (Tassa et al., 2018)) with task-irrelevant visual distractors in backgrounds. The backgrounds are replaced with natural videos from the Kinetics dataset (Kay et al., 2017) (middle column in Figure 2). In Appedix C.1, we also show comparison between DRIBO and SOTAs on DMC environments without the background distractors (left column in Figure 2).

For generalization, we present results on Procgen (Cobbe et al., 2020) which provides different levels of the same game to test how well agents generalize to unseen levels. We use single-step observations to train DRIBO without assuming that observation at each timestep provides full observability of the underlying dynamics. By contrast, current SOTA approaches require the use of consecutive observations [1] to capture predictive properties of the underlying states.

We also conduct careful ablations analysis to show that the benefit of DRIBO is due primarily to learning from the temporal structure of RL and the DRIBO loss.

For the DMC suite, all agents are built on top of SAC. For the Procgen suite, we augment PPO, a RL baseline for Procgen, with DRIBO. Implementation details are in Appendix D.

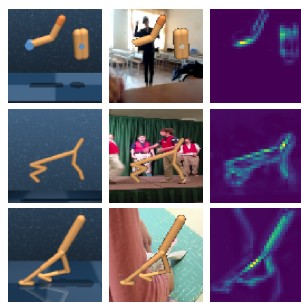

Figure 2: Left: DMC observations without visual distractors. Middle: observations with natural videos as backgrounds. Right: spatial attention maps of encoders for the middle images.

## 5.1 EFFECTIVENESS AND ROBUSTNESS

We compare DRIBO against several SOTA methods. The first is RAD (Laskin et al., 2020b), a recent method that uses augmented data to train pixel-based policies on DMC benchmarks. The second is SLAC (Lee et al., 2020a), a SOTA representation learning method for RL that learns a dynamic model using a reconstruction loss. The third is CURL (Laskin et al., 2020a), an approach that leverages contrastive learning to maximize the mutual information between representations of augmented versions of the same observation but does not distinguish between relevant and irrelevant features. The fourth is DBC (Zhang et al., 2021) which shares a similar goal with DRIBO. DBC learns an invariant representation based on bisimulation metrics without requiring reconstruction. Finally, we compare with PI-SAC (Lee et al., 2020b) which leverages the Predictive Information to compress away task-irrelevant information with CEB. We apply *random crop+grayscale* to obtain the augmented data for RAD which achieves the best performance in (Laskin et al., 2020b). For CURL and DRIBO, we apply *random crop* to obtain augmented data and multi-view observations.

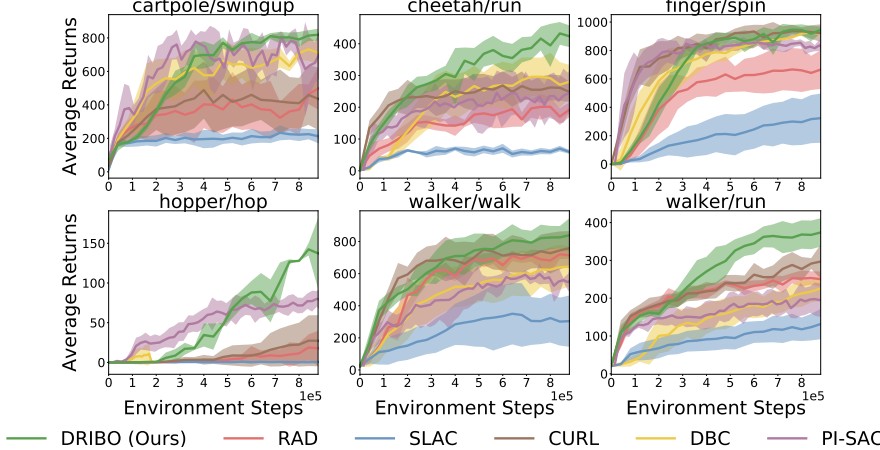

Figure 3: Results for DMC over 5 seeds with one standard error shaded in the natural video setting.

**Natural Video Setting.** To investigate the effectiveness and robustness of RL agents in DMC environments, we introduce high-dimensional visual distractors by using natural videos from the Kinetics dataset (Kay et al., 2017) as backgrounds (Zhang et al., 2018a) (Figure 2: middle column). We use the class of "arranging flowers" videos to replace the background in training. During testing, we use the test set from the Kinetics dataset to replace the background, which contains videos from various different classes. Note that a single run of the DMC task may have *multiple videos playing sequentially* in the background. See Appendix C.4 for snapshots under the natural video setting.

---

[1]All other methods compared in this paper use stack frames of 3 consecutive observations.

In Figure 2, spatial attention maps (Zagoruyko & Komodakis, 2017) of the trained DRIBO encoder demonstrate that DRIBO trains agents to focus on the robot body while ignoring irrelevant scene details in the background. Figure 3 shows that DRIBO performs substantially better than RAD, SLAC and CURL which do not discard task-irrelevant information explicitly. Compared to PI-SAC and DBC which are recent state-of-art methods aimed at learning representations invariant to task-irrelevant information, DRIBO outperforms them consistently – DRIBO achieves on average 30% and at the maximum 77% higher returns at 88e4 steps compared to the second best performing method.

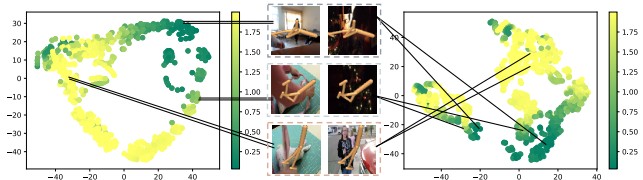

Figure 4: t-SNE of latent spaces learned with DRIBO (left t-SNE) and CURL (right t-SNE). We color-code the embedded points with reward values (higher value yellow, lower value green). Each pair of solid lines indicates the corresponding embedded points for observations with an identical foreground but different backgrounds. DRIBO learns representations that are neighboring in the embedding space with similar reward values. This property holds even if the backgrounds are drastically different (see middle images). By contrast, CURL maps the same image pairs to points far away from each other in the embedding space.

**Visualization.** We visualize the representations learned by DRIBO and CURL with t-SNE (Van der Maaten & Hinton, 2008). Figure 4 shows that even when the background looks drastically different, DRIBO learns to disregard irrelevant information and maps observations with similar robot configurations to the neighborhoods of one another. The color code represents value of the reward for each representation. The rewards can be viewed as task-relevant signals provided by the environments. DRIBO learns representations that are close in the latent space with similar reward values.

## 5.2 ABLATIONS

**Temporal Structure of RL.** To investigate whether DRIBO captures the temporal structure of RL, we conducted further experiments on DRIBO agents trained using sequences of different lengths under the natural video setting. Longer sequences carry more temporal information for DRIBO to learn. By default, we train DRIBO with sequences of length 32. In this ablation study, we present results of DRIBO trained using sequences of lengths 3, 6 and 16 respectively in Figure 5. Using sequences of length 3 is similar to stacking 3 consecutive frames which is a common choice for training in DMC.

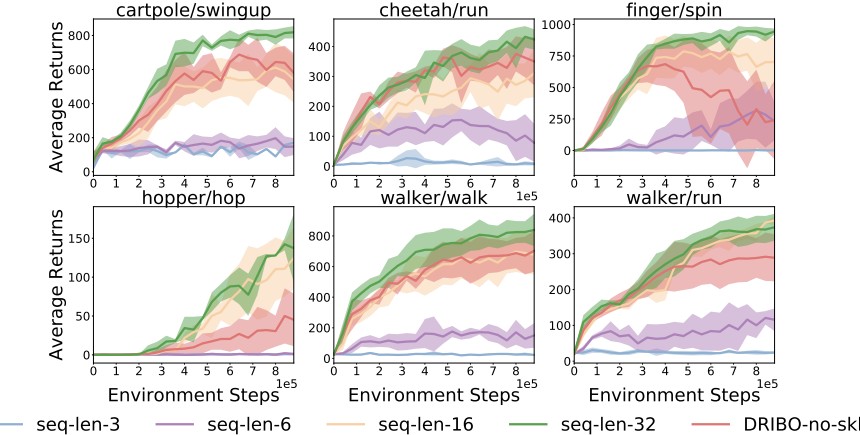

Figure 5: DRIBO achieves better performance by capturing the temporal structure of RL from longer training sequences. Compressing away task-irrelevant information using the SKL term in DRIBO loss improves performance when the architecture choice and training configurations are the same. We perform 5 runs for each method under the natural video setting. More results are in Appendix C.2.

Using sequences of length 6 is similar to the design choice made in PI-SAC (3 steps for the past and 3 steps for the future). We also include results on using 16-step sequential observations to investigate how DRIBO's performance changes as the length of the sequences increases. Theoretically, the DRIBO loss provides a lower bound on the mutual information between sequences of observations and sequences of (latent) representations with the same length. It can be observed that DRIBO performs significantly better when training with longer sequences.

**Learning Objective.** In the DRIBO loss, we use the SKL term (second term in Eq. (7)) to compress away task-irrelevant information. Figure 5 studies the effect of removing the SKL term in the DRIBO loss (annotated as DRIBO-no-skl). The objective without the SKL term is equivalent to an InfoMax objective. We use identical training configurations (e.g. sequence length of 32) for DRIBO-no-skl and DRIBO. The only difference is whether the SKL term is included in the learning objective. Figure 5 shows that DRIBO outperforms DRIBO-no-skl substantially in the natural video setting.

## 5.3 GENERALIZATION

The natural video setting of DMC is suitable for benchmarking robustness to high-dimensional visual distractors. However, the task-relevant information and the task difficulties are unchanged. Thus, we use the ProcGen suite to investigate the generalization capabilities of DRIBO. For each game, agents are trained on the first 200 levels, and evaluated w.r.t. their zero-shot performance averaged over unseen levels during testing. Unseen levels typically have different backgrounds or layouts, which are relatively easy for humans to adapt to but challenging for RL agents.

We compare DRIBO with recent methods that incorporate data augmentation. In addition to comparing with RAD, we compare DRIBO with DrAC (Raileanu et al., 2020) which applies two regularization terms for policy and value function using augmented data. UCB-DrAC is built on top of DrAC, which automatically selects the best type of data augmentation for DrAC. For RAD and DrAC, we use the best reported augmentation types for different environments. DRIBO selects the same augmentation types except for a few games. The details can be found in Appendix D. We also compare the Procgen results with DAAC (Raileanu & Fergus, 2021) and IDAAC (Raileanu & Fergus, 2021), two state-of-art methods on the Procgen suite that do not apply data augmentation. DAAC decouples the learning of the policy and value function in RL to improve the generalization of RL. IDAAC is built on top of DAAC by adding an auxiliary loss based on an adversarial framework.

Table 1 shows that DRIBO achieves higher averaged testing returns compared to the PPO baseline and other augmentation-based methods. The few environments, in which our approach does not outperform the other augmentation-based methods, share the commonality that task-relevant layouts remain static throughout the same run of the game. Since the current version of DRIBO only considers the mutual information between the complete input and the encoder output (global MI (Hjelm et al., 2019)), it may fail to capture local features. The representations for a sequence of observations within the same run of the game are treated as

Table 1: Procgen returns on test levels after training on 25M environment steps. The mean and standard deviation are computed over 10 seeds.

| Env | PPO | RAD | DrAC | UCB-DrAC | DAAC | IDAAC | DRIBO |
|---|---|---|---|---|---|---|---|
| BigFish | 4.0 ± 1.2 | 9.9 ± 1.7 | 8.7 ± 1.4 | 9.7 ± 1.0 | 17.8 ± 1.4 | **18.5 ± 1.2** | 10.9 ± 1.6 |
| StarPilot | 24.7 ± 3.4 | 33.4 ± 5.1 | 29.5 ± 5.4 | 30.2 ± 2.8 | 36.4 ± 2.8 | **37.0 ± 2.3** | 36.5 ± 3.0 |
| FruitBot | 26.7 ± 0.8 | 27.3 ± 1.8 | 28.2 ± 0.8 | 28.3 ± 0.9 | 28.6 ± 0.6 | 27.9 ± 0.5 | **30.8 ± 0.8** |
| BossFight | 7.7 ± 1.0 | 7.9 ± 0.6 | 7.5 ± 0.8 | 8.3 ± 0.8 | 9.6 ± 0.5 | 9.8 ± 0.6 | **12.0 ± 0.5** |
| Ninja | 5.9 ± 0.7 | 6.9 ± 0.8 | 7.0 ± 0.4 | 6.9 ± 0.6 | 6.8 ± 0.4 | 6.8 ± 0.4 | **9.7 ± 0.7** |
| Plunder | 5.0 ± 0.5 | 8.5 ± 1.2 | 9.5 ± 1.0 | 8.9 ± 1.0 | 20.7 ± 3.3 | **23.3 ± 1.4** | 5.8 ± 1.0 |
| CaveFlyer | 5.1 ± 0.9 | 5.1 ± 0.6 | 6.3 ± 0.8 | 5.3 ± 0.9 | 4.6 ± 0.2 | 5.0 ± 0.6 | **7.5 ± 1.0** |
| CoinRun | 8.5 ± 0.5 | 9.0 ± 0.8 | 8.8 ± 0.2 | 8.5 ± 0.6 | 9.2 ± 0.2 | **9.4 ± 0.1** | 9.2 ± 0.7 |
| Jumper | 5.8 ± 0.5 | 6.5 ± 0.6 | 6.6 ± 0.4 | 6.4 ± 0.6 | 6.5 ± 0.4 | 6.3 ± 0.2 | **8.4 ± 1.6** |
| Chaser | 5.0 ± 0.8 | 5.9 ± 1.0 | 5.7 ± 0.6 | 6.7 ± 0.6 | 6.6 ± 1.2 | **6.8 ± 1.0** | 4.8 ± 0.8 |
| Climber | 5.7 ± 0.8 | 6.9 ± 0.8 | 7.1 ± 0.7 | 6.5 ± 0.8 | 7.8 ± 0.2 | **8.3 ± 0.4** | 8.1 ± 1.6 |
| DodgeBall | **11.7 ± 0.3** | 2.8 ± 0.7 | 4.3 ± 0.8 | 4.7 ± 0.7 | 3.3 ± 0.5 | 3.3 ± 0.3 | 3.8 ± 0.9 |
| Heist | 2.4 ± 0.5 | 4.1 ± 1.0 | 4.0 ± 0.8 | 4.0 ± 0.7 | 3.3 ± 0.2 | 3.5 ± 0.2 | **7.7 ± 1.6** |
| Leaper | 4.9 ± 0.7 | 4.3 ± 1.0 | 5.3 ± 1.1 | 5.0 ± 0.3 | 7.3 ± 1.1 | **7.7 ± 1.0** | 5.3 ± 1.5 |
| Maze | 5.7 ± 0.6 | 6.1 ± 1.0 | 6.6 ± 0.8 | 6.3 ± 0.6 | 5.5 ± 0.2 | 5.6 ± 0.3 | **8.5 ± 1.6** |
| Miner | 8.5 ± 0.5 | 9.4 ± 1.2 | **9.8 ± 0.6** | 9.7 ± 0.7 | 8.6 ± 0.9 | 9.5 ± 0.4 | **9.8 ± 0.9** |

globally negative pairs in DRIBO but they may be locally positive pairs. Thus, the performance of DRIBO can be further improved by considering local features (e.g. positions of the layouts) shared between representations as positive pairs in the mutual information estimation. We leave this investigation to future work. DRIBO also outperforms DAAC and IDAAC in *9 of the 16 games*.

## 6 CONCLUSION

We introduce a novel robust representation learning approach based on the multi-view information bottleneck principle for RL problems. Our experimental results show that (1) DRIBO learns representations that are robust against task-irrelevant distractions and boosts the RL agent's performance even when complex visual distractors are introduced, and (2) DRIBO improves generalization performance compared to well-established baselines on the large-scale Procgen benchmarks.

**Reproducibility Statement.** The implementation code can be found in Supplementary Material.zip, and we will also release it on GitHub once the paper is published. All datasets we use are public. In addition, we also provide detailed experiment parameters in the Appendix D.

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

APPENDIX

## A  THEOREMS AND PROOFS

In this section, we first list properties of the mutual information we use in our proof. For any random variables $X, Y$ and $Z$.

**(P.1)** Positivity:
$$I(X;Y) \geq 0, I(X;Y|Z) \geq 0$$

**(P.2)** Chain rule:
$$I(XY;Z) = I(Y;Z) + I(X;Z|Y)$$

**(P.3)** Chain rule (Multivariate Mutual Information):
$$I(X;Y;Z) = I(Y;Z) - I(Y;Z|X)$$

**(P.4)** Entropy and Mutual Information:
$$I(X;Y) = H(X) - H(X|Y)$$

**(P.5)** Chain rule for Entropy:
$$H(X_1, X_2, \ldots, X_n) = \sum_{i=1}^{n} H(X_i|X_{i-1}, \ldots, X_1)$$

**(P.6)** The mutual information among three variables is bounded by:
$$-\min\{I(X;Y|Z), I(X;Z|Y), I(Y;Z|X)\} \leq I(X;Y;Z) \leq \min\{I(X;Y), I(X;Z), I(Y;Z)\}$$

### A.1  THEOREM 1

We first show that $I(S_{1:T}; O_{1:T}|A_{1:T})$ is a lower bound of $I(S_{1:T}; O_{1:T})$.

$$I(S_{1:T}; O_{1:T}|A_{1:T}) \overset{\text{(P.3)}}{=} I(O_{1:T}; S_{1:T}) - I(S_{1:T}; O_{1:T}; A_{1:T})$$

Since $S_{1:T}$ are representations of $O_{1:T}$, we have $I(S_{1:T}; A_{1:T}|O_{1:T}) = 0$. With **(P.1)** and **(P.6)**, we have that:

$$I(S_{1:T}; O_{1:T}; A_{1:T}) \geq 0$$

Thus, we have:

$$I(O_{1:T}; S_{1:T}) \geq I(S_{1:T}; O_{1:T}|A_{1:T})$$

Furthermore, since $I(S_{1:T}; O_{1:T}; A_{1:T})$ is lower bounded, maximizing $I(S_{1:T}; O_{1:T}|A_{1:T})$ also maximizes $I(O_{1:T}; S_{1:T})$.

**Theorem A.1.** *Let $O_{1:T}$ be the observation sequence obtained by executing action sequence $A_{1:T}$. If $S_{1:T}$ is a sequence of sufficient representations for $O_{1:T}$, then we have:*

$$I(S_{1:T}; O_{1:T}|A_{1:T}) \geq \sum_{t=1}^{T} I(S_t; O_t|S_{t-1}, A_{t-1}) \tag{8}$$

*Proof.* We indicate the property we use for each step of the derivation below.

$$I(S_{1:T}; O_{1:T}|A_{1:T})$$

$$\stackrel{\text{(P.4)}}{=} H(S_{1:T}|A_{1:T}) - H(S_{1:T}|O_{1:T}, A_{1:T})$$

$$\stackrel{\text{(P.5)}}{=} \sum_t \left( H(S_t|A_{1:T}, S_{1:t-1}) - H(S_t|A_{1:T}, O_{1:T}, S_{1:t-1}) \right)$$

$$\stackrel{\text{(P.4)}}{=} \sum_t I(S_t; O_{1:T}|A_{1:T}, S_{1:t-1})$$

$$\stackrel{\text{(P.4)}}{=} \sum_t \left( H(O_{1:T}|A_{1:T}, S_{1:t-1}) - H(O_{1:T}|S_t, A_{1:T}, S_{1:t-1}) \right)$$

$$\stackrel{\text{(P.5)}}{=} \sum_t \sum_\tau \big( H(O_\tau|A_{1:T}, S_{1:t-1}, O_{1:\tau-1})$$
$$- H(O_\tau|S_t, A_{1:T}, S_{1:t-1}, O_{1:\tau-1}) \big)$$

$$\stackrel{\text{(P.4)}}{=} \sum_t \sum_\tau I(S_t; O_\tau|A_{1:T}, S_{1:t-1}, O_{1:\tau-1})$$

$$\stackrel{\text{(P.1)}}{\geq} \sum_t I(S_t; O_t|A_{1:T}, S_{1:t-1}, O_{1:t-1})$$

$$= \sum_t I(S_t; O_t|S_{t-1}, A_{t-1})$$

Here, we provide a formal proof for the last step of the derivation above. Let $\tau$ represent $\boldsymbol{a}_{1:T}, \boldsymbol{s}_{1:t-1}, \boldsymbol{o}_{1:t-1}$. We have

$$I(S_t; O_t|A_{1:T}, S_{1:t-1}, O_{1:t-1})$$
$$= \int_\tau \int_{\boldsymbol{s}_t} \int_{\boldsymbol{o}_t} p(\tau) p(\boldsymbol{s}_t, \boldsymbol{o}_t|\tau) \log \left[ \frac{p(\boldsymbol{s}_t, \boldsymbol{o}_t|\boldsymbol{a}_{1:T}, \boldsymbol{s}_{1:t-1}, \boldsymbol{o}_{1:t-1})}{p(\boldsymbol{s}_t|\boldsymbol{a}_{1:T}, \boldsymbol{s}_{1:t-1}, \boldsymbol{o}_{1:t-1}) p(\boldsymbol{o}_t|\boldsymbol{a}_{1:T}, \boldsymbol{s}_{1:t-1}, \boldsymbol{o}_{1:t-1})} \right] d\boldsymbol{s}_t d\boldsymbol{o}_t d\tau$$

With the density of multi-view trajectories Eq. (1), we can observe that $\boldsymbol{o}_t$ and $\boldsymbol{s}_t$ are generated by $p(\boldsymbol{o}_t|\boldsymbol{s}_t)p(\boldsymbol{s}_t|\boldsymbol{s}_{t-1}, \boldsymbol{a}_{t-1}) = p(\boldsymbol{s}_t, \boldsymbol{o}_t|\boldsymbol{s}_{t-1}, \boldsymbol{a}_{t-1})$. Thus, we have:

$$p(\boldsymbol{s}_t, \boldsymbol{o}_t|\boldsymbol{a}_{1:T}, \boldsymbol{s}_{1:t-1}, \boldsymbol{o}_{1:t-1}) = p(\boldsymbol{s}_t, \boldsymbol{o}_t|\boldsymbol{s}_{t-1}, \boldsymbol{a}_{t-1})$$
$$p(\boldsymbol{s}_t|\boldsymbol{a}_{1:T}, \boldsymbol{s}_{1:t-1}, \boldsymbol{o}_{1:t-1}) = p(\boldsymbol{s}_t|\boldsymbol{s}_{t-1}, \boldsymbol{a}_{t-1})$$
$$p(\boldsymbol{o}_t|\boldsymbol{a}_{1:T}, \boldsymbol{s}_{1:t-1}, \boldsymbol{o}_{1:t-1}) = p(\boldsymbol{o}_t|\boldsymbol{s}_{t-1}, \boldsymbol{a}_{t-1})$$

This in turn implies that:

$$I(S_t; O_t|A_{1:T}, S_{1:t-1}, O_{1:t-1}) = I(S_t; O_t|S_{t-1}, A_{t-1})$$

$\square$

As a result, we have a lower bound of $I(S_{1:T}^{(1)}; O_{1:T}^{(1)}|A_{1:T})$:

$$I(S_{1:T}^{(1)}; O_{1:T}^{(1)}|A_{1:T}) \geq \sum_t \left( I(S_t^{(1)}; O_t^{(1)}|S_{t-1}^{(1)}, A_{t-1}, O_t^{(2)}) + I(O_t^{(2)}; S_t^{(1)}|S_{t-1}^{(1)}, A_{t-1}) \right)$$

With the information bottleneck principle, we have losses in Eq. (5) and Eq. (6) to compress away task-irrelevant information in representations while maximizing the mutual information between representations and observations.

## A.2 SUFFICIENT REPRESENTATIONS IN RL

In this section, we first present the sufficiency condition for sequential data. Then, we prove that if the sufficiency condition on the sequential data holds, then the sufficiency condition on each corresponding individual representation and observation holds as well.

**Theorem A.2.** *Let $O_{1:T}$ and $A^*_{1:T}$ be random variables with joint distribution $p(\boldsymbol{o}_{1:T}, \boldsymbol{a}^*_{1:T})$. Let $S_{1:T}$ be the representation of $O_{1:T}$, then $S_{1:T}$ is sufficient for $A^*_{1:T}$ if and only if $I(O_{1:T}; A^*_{1:T}) = I(S_{1:T}; A^*_{1:T})$. Also, $S_t$ is a sufficient representation of $O_t$ since $I(O_t; A^*_t | S_t, S_{t-1}, A_{t-1}) = 0$.*

*Hypothesis:*

*(H.1) $S_{1:T}$ is a sequence of sufficient representations for $O_{1:T}$:*

$$I(O_{1:T}; A^*_{1:T} | S_{1:T}) = 0$$

*Proof.*

$$
\begin{aligned}
&I(O_{1:T}; A^*_{1:T} | S_{1:T}) \\
&\overset{\text{(P.3)}}{=} I(O_{1:T}; A^*_{1:T}) - I(O_{1:T}; A^*_{1:T}; S_{1:T}) \\
&\overset{\text{(P.3)}}{=} I(O_{1:T}; A^*_{1:T}) - I(A^*_{1:T}; S_{1:T}) - I(A^*_{1:T}; S_{1:T} | O_{1:T})
\end{aligned}
$$

With $S_{1:T}$ as a representation of $O_{1:T}$, we have $I(S_{1:T}; A^*_{1:T} | O_{1:T}) = 0$. The reason is that $O_{1:T}$ shares the same level of information as $A^*_{1:T}$ and $S_{1:T}$. Then,

$$I(O_{1:T}; A^*_{1:T} | S_{1:T}) = I(O_{1:T}; A^*_{1:T}) - I(A^*_{1:T}; S_{1:T}) \tag{9}$$

So the sufficiency condition $I(O_{1:T}; A^*_{1:T} | S_{1:T}) = 0$ holds if and only if $I(O_{1:T}; A^*_{1:T}) = I(A^*_{1:T}; S_{1:T})$.

We factorize the mutual information between sequential observations and optimal actions

$$
\begin{aligned}
&I(O_{1:t}; A^*_{1:t}) \\
&\overset{\text{(P.2)}}{=} I(O_t; A^*_{1:t} | O_{1:t-1}) + I(O_{1:t-1}; A^*_{1:t}) \\
&\overset{\text{(H.1)}}{=} I(O_t; A^*_{1:t} | O_{1:t-1}) + I(S_{1:t-1}; A^*_{1:t})
\end{aligned}
$$

$$
\begin{aligned}
&I(S_{1:t}; A^*_{1:t}) \\
&\overset{\text{(P.2)}}{=} I(S_t; A^*_{1:t} | S_{1:t-1}) + I(S_{1:t-1}; A^*_{1:t}) \\
&\overset{\text{(H.1)}}{=} I(S_t; A^*_{1:t} | S_{1:t-1}) + I(O_{1:t-1}; A^*_{1:t})
\end{aligned}
$$

Then we obtain the following relation:

$$I(O_t; A^*_{1:t} | O_{1:t-1}) = I(S_t; A^*_{1:t} | S_{1:t-1}) \tag{10}$$

We also have

$$I(O_t; A_{1:t}^* | O_{1:t-1})$$

$$\overset{\text{(P.2)}}{=} I(O_{1:t}; A_{1:t}^*) - I(O_{1:t-1}; A_{1:t}^*)$$

$$\overset{\text{(P.2)}}{=} I(O_{1:t-1}; A_{1:t}^* | O_t) + I(O_t; A_{1:t}^*)$$
$$- I(O_{1:t-1}; A_{1:t}^*)$$

$$\overset{\text{(H.1)}}{=} I(S_{1:t-1}; A_{1:t}^* | O_t) + I(O_t; A_{1:t}^*)$$
$$- I(S_{1:t-1}; A_{1:t}^*)$$

$$\overset{\text{(P.2)}}{=} I(O_t S_{1:t-1}; A_{1:t}^*) - I(S_{1:t-1}; A_{1:t}^*)$$

$$\overset{\text{(P.2)}}{=} I(O_t; A_{1:t}^* | S_{1:t-1})$$

$$\overset{\text{Eq. (10)}}{=} I(S_t; A_{1:t}^* | S_{1:t-1})$$

$$\overset{\text{(P.2)}}{\Longleftrightarrow}$$

$$I(O_t; A_t^* | A_{1:t-1}^*, S_{1:t-1}) + I(O_t; A_{1:t-1}^* | S_{1:t-1})$$
$$= I(S_t; A_t^* | A_{1:t-1}^*, S_{1:t-1}) + I(S_t; A_{1:t-1}^* | S_{1:t-1})$$

$$\overset{\text{Eq. (10)}}{\Longleftrightarrow}$$

$$I(O_t; A_t^* | A_{1:t-1}^*, S_{1:t-1}) = I(S_t; A_t^* | A_{1:t-1}^*, S_{1:t-1})$$

$$\overset{\text{Eq. (9)}}{\Longleftrightarrow}$$

$$I(O_t, A_t^* | S_t, S_{1:t-1}, A_{1:t-1}^*) = 0$$

With the above derivation and Markov property, we have $I(O_t; A_t^* | S_t, S_{t-1}, A_{t-1}) = 0$. We can generalize $A_{t-1}^*$ to any $A_{t-1}$ by assuming $A_t^*$ as the optimal action for state $S_t$ whose last timestep state-action pair is $(S_{t-1}, A_{t-1})$. Thus, we have $S_t$ is a sufficient representation for $O_t$ if and only if $S_{1:T}$ is a sufficient representation of $O_{1:T}$. □

## A.3 MULTI-VIEW REDUNDANCY AND SUFFICIENCY

**Proposition A.1.** $O_{1:T}^{(1)}$ is a redundant view with respect to $O_{1:T}^{(2)}$ to obtain $A_{1:T}^*$ if only if $I(O_{1:T}^{(1)}; A_{1:T}^* | O_{1:T}^{(2)}) = 0$. Any representation $S_{1:T}^{(1)}$ of $O_{1:T}^{(1)}$ that is sufficient for $O_{1:T}^{(2)}$ is also sufficient for $A_{1:T}^*$.

*Proof.* See proof of Proposition B.3 in the MIB paper (Federici et al., 2020). □

**Corollary A.1.** Let $O_{1:T}^{(1)}$ and $O_{1:T}^{(2)}$ be two mutually redundant views for $A_{1:T}^*$. Let $S_{1:T}^{(1)}$ be a representation of $O_{1:T}^{(1)}$. If $S_{1:T}^{(1)}$ is sufficient for $O_{1:T}^{(2)}$, $S_t^{(1)}$ can derive $A_t^*$ as the joint observation of the two views $(I(O_t^{(1)} O_t^{(2)}; A_t^* | S_{t-1}, A_{t-1}) = I(S_t^{(1)}; A_t^* | S_{t-1}, A_{t-1}))$, where $S_{t-1}$ is any sufficient representation at timestep $t-1$.

*Proof.* For the sequential data, see proof of Corollary B.2.1 in the MIB paper (Federici et al., 2020) to prove

$$I(O_{1:T}^{(1)} O_{1:T}^{(2)}; A_{1:T}^*) = I(S_{1:T}^{(1)}; A_{1:T}^*)$$

According to Theorem A.2, if $S_{1:T}^{(1)}$ is a sufficient representation of $O_{1:T}^{(2)}$, $S_t^{(1)}$ is a sufficient representation of $O_t^{(2)}$. Similar to proof on sequential data, we can use Corollary B.2.1 in the MIB paper (Federici et al., 2020) to show that

$$I(O_t^{(1)} O_t^{(2)}; A_t^* | S_{t-1}, A_{t-1}) = I(S_t^{(1)}; A_t^* | S_{t-1}, A_{t-1})$$

□

**Theorem A.3.** *Let the two views $\boldsymbol{o}_{1:T}^{(1)}$ and $\boldsymbol{o}_{1:T}^{(2)}$ of observation $\boldsymbol{o}_{1:T}$ are obtained by data augmentation transformation sequences $t_{1:T}^{(1)}$ and $t_{1:T}^{(2)}$ respectively ($\boldsymbol{o}_{1:T}^{(1)}{=}t_{1:T}^{(1)}(\boldsymbol{o}_{1:T})$ and $\boldsymbol{o}_{1:T}^{(2)}{=}t_{1:T}^{(2)}(\boldsymbol{o}_{1:T})$). We abuse the notation $t$ for simplicity to represent $t_{1:T}^{(1)}(O_{1:T})$ and $t_{1:T}^{(2)}(O_{1:T})$ as random variables for augmented observations. Whenever $I(t_{1:T}^{(1)}(O_{1:T}); A_{1:T}^{*}){=}I(t_{1:T}^{(2)}(O_{1:T}); A_{1:T}^{*}){=}I(O_{1:T}; A_{1:T}^{*})$, the two views $O_{1:T}^{(1)}$ and $O_{1:T}^{(2)}$ must be mutually redundant for $A_{1:T}^{*}$. Besides, the two views $O_t^{(1)}$ and $O_t^{(2)}$ must be mutually redundant for $A_t^{*}$.*

*Proof.* Let $\boldsymbol{s}_{1:T}$ be a sufficient representation for both original and multi-view observations. We first factorize the mutual information and refer A.2 as Theorem A.2.

$$I(t_{1:t}^{(1)}(O_{1:t}); A_{1:t}^{*}) = I(O_{1:t}^{(1)}; A_{1:t}^{*})$$
$$\overset{\text{(P.2)}}{=} I(O_t^{(1)}; A_{1:t}^{*}|O_{1:t-1}^{(1)}) + I(O_{1:t-1}^{(1)}; A_{1:t}^{*})$$
$$\overset{A.2}{=} I(O_t^{(1)}; A_{1:t}^{*}|S_{1:t-1}) + I(S_{1:t-1}; A_{1:t}^{*})$$

$$I(t_{1:t}^{(2)}(O_{1:t}); A_{1:t}^{*}) = I(O_{1:t}^{(2)}; A_{1:t}^{*})$$
$$\overset{\text{(P.2)}}{=} I(O_t^{(2)}; A_{1:t}^{*}|O_{1:t-1}^{(2)}) + I(O_{1:t-1}^{(2)}; A_{1:t}^{*})$$
$$\overset{A.2}{=} I(O_t^{(2)}; A_{1:t}^{*}|S_{1:t-1}) + I(S_{1:t-1}; A_{1:t}^{*})$$

$$I(O_{1:t}; A_{1:t}^{*}) = I(O_{1:t}; A_{1:t}^{*})$$
$$\overset{\text{(P.2)}}{=} I(O_t; A_{1:t}^{*}|O_{1:t-1}) + I(O_{1:t-1}; A_{1:t}^{*})$$
$$\overset{A.2}{=} I(O_t; A_{1:t}^{*}|S_{1:t-1}) + I(S_{1:t-1}; A_{1:t}^{*})$$

Then, we have the following equality

$$I(O_t^{(1)}; A_{1:t}^{*}|S_{1:t-1}){=}I(O_t^{(2)}; A_{1:t}^{*}|S_{1:t-1}){=}I(O_t; A_{1:t}^{*}|S_{1:t-1})$$

Similar as derivation in Theorem A.2

$$I(O_t^{(1)}; A_{1:t}^{*}|S_{1:t-1})$$
$$\overset{\text{(P.2)}}{=} I(O_t^{(1)}; A_t^{*}|A_{1:t-1}^{*}, S_{1:t-1}) + I(O_t^{(1)}; A_{1:t-1}^{*}|S_{1:t-1})$$
$$\overset{\text{Eq. (10)}}{=} I(O_t^{(1)}; A_t^{*}|A_{1:t-1}^{*}, S_{1:t-1}) + I(S_t; A_{1:t-1}^{*}|S_{1:t-1})$$

We apply the same derivation for $\boldsymbol{o}^{(2)}$ and $\boldsymbol{o}$, we have the following with Markov property

$$I(t_t^{(1)}(O_t); A_t^{*}|S_{t-1}, A_{t-1})$$
$$=I(t_t^{(2)}(O_t); A_t^{*}|S_{t-1}, A_{t-1})$$
$$=I(O_t; A_t^{*}|S_{t-1}, A_{t-1})$$

We show that the condition on sequential data can be expressed at each timestep with the similar form. See proof of Proposition B.4 in the MIB paper (Federici et al., 2020) for mutual redundancy between sequential views and individual pairs of views. □

**Theorem A.4.** *Suppose the mutually redundant condition holds, i.e. $I(t_{1:T}^{(1)}(O_{1:T}); A_{1:T}^{*}){=}I(t_{1:T}^{(2)}(O_{1:T}); A_{1:T}^{*}){=}I(O_{1:T}; A_{1:T}^{*})$. If $S_{1:T}^{(1)}$ is a sufficient representation for $t_{1:T}^{(2)}(O_{1:T})$ then $I(O_t; A_t^{*}|S_{t-1}, A_{t-1}) = I(S_t^{(1)}; A_t^{*}|S_{t-1}, A_{t-1})$.*

*Proof.* Since $t_{1:T}^{(1)}(O_{1:T})$ is redundant for $t_{1:T}^{(2)}(O_{1:T})$ (Theorem A.3), any representation $S_t^{(1)}$ of $t_{1:T}^{(1)}(O_{1:T})$ that is sufficient for $t_{1:T}^{(2)}(O_{1:T})$ must also be sufficient for $A_t^{*}$ (Theorem A.2 and Proposition A.1). Using Theorem A.2 we have $I(S_t^{(1)}; A_t^{*}|S_{t-1}, A_{t-1}){=}I(t_t^{(1)}(O_t); A_t^{*}|S_{t-1}, A_{t-1})$. With $I(t_t^{(1)}(O_t); A_t^{*}|S_{t-1}, A_{t-1}) = I(O_t; A_t^{*}|S_{t-1}, A_{t-1})$, we conclude $I(O_t; A_t^{*}|S_{t-1}, A_{t-1}) = I(S_t^{(1)}; A_t^{*}|S_{t-1}, A_{t-1})$. □

We finally show the proposition for the Multi-Information Bottleneck principle in RL with the generalization of sufficiency and mutually redundancy condition from sequential data to each individual pairs of data.

**Proposition A.2.** Let $O_t^{(1)}$ and $O_t^{(2)}$ be mutually redundant views for $A_t^*$ that share only optimal action information. Then a sufficient representation of $S_t^{(1)}$ of $O_t^1$ for $O_t^{(2)}$ that is minimal for $O_t^{(2)}$ is also a minimal representation for $A_t^*$.

*Proof.* See proof of Proposition E.1 in the MIB paper (Federici et al., 2020). □

# B    DRIBO LOSS COMPUTATION

We consider the average of the information bottleneck losses from the two views.

$$\mathcal{L}_{\frac{1+2}{2}} \tag{11}$$

$$=\frac{I(S_t^{(1)};O_t^{(1)}|S_{t-1}^{(1)},A_{t-1},O_t^{(2)})+I(S_t^{(2)};O_t^{(2)}|S_{t-1}^{(2)},A_{t-1},O_t^{(1)})}{2}$$

$$-\frac{\lambda_1 I(S_t^{(1)};O_t^{(2)}|S_{t-1}^{(1)},A_{t-1})+\lambda_2 I(S_t^{(2)};O_t^{(1)}|S_{t-1}^{(2)},A_{t-1})}{2} \tag{12}$$

Consider $\boldsymbol{s}_t^{(1)}$ and $\boldsymbol{s}_t^{(2)}$ on the same domain $\mathbb{S}$, $I(S_t^{(1)};O_t^{(1)}|S_{t-1}^{(1)},A_{t-1},O_t^{(2)})$ can be expressed as:

$$I(S_t^{(1)};O_t^{(1)}|S_{t-1}^{(1)},A_{t-1},O_t^{(2)})$$

$$=\mathbb{E}\left[\log\frac{p_\theta(\boldsymbol{s}_t^{(1)}|\boldsymbol{o}_t^{(1)},\boldsymbol{s}_{t-1}^{(1)},\boldsymbol{a}_{t-1})}{p_\theta(\boldsymbol{s}_t^{(1)}|\boldsymbol{o}_t^{(2)},\boldsymbol{s}_{t-1}^{(1)},\boldsymbol{a}_{t-1})}\right]$$

$$=\mathbb{E}\left[\log\frac{p_\theta(\boldsymbol{s}_t^{(1)}|\boldsymbol{o}_t^{(1)},\boldsymbol{s}_{t-1}^{(1)},\boldsymbol{a}_{t-1})}{p_\theta(\boldsymbol{s}_t^{(2)}|\boldsymbol{o}_t^{(2)},\boldsymbol{s}_{t-1}^{(2)},\boldsymbol{a}_{t-1})}\frac{p_\theta(\boldsymbol{s}_t^{(2)}|\boldsymbol{o}_t^{(2)},\boldsymbol{s}_{t-1}^{(2)},\boldsymbol{a}_{t-1})}{p_\theta(\boldsymbol{s}_t^{(1)}|\boldsymbol{o}_t^{(2)},\boldsymbol{s}_{t-1}^{(1)},\boldsymbol{a}_{t-1})}\right]$$

$$=D_{\mathrm{KL}}(p_\theta(\boldsymbol{s}_t^{(1)}|\boldsymbol{o}_t^{(1)},\boldsymbol{s}_{t-1}^{(1)},\boldsymbol{a}_{t-1})||p_\theta(\boldsymbol{s}_t^{(2)}|\boldsymbol{o}_t^{(2)},\boldsymbol{s}_{t-1}^{(2)},\boldsymbol{a}_{t-1}))$$

$$-D_{\mathrm{KL}}(p_\theta(\boldsymbol{s}_t^{(1)}|\boldsymbol{o}_t^{(2)},\boldsymbol{s}_{t-1}^{(1)},\boldsymbol{a}_{t-1})||p_\theta(\boldsymbol{s}_t^{(2)}|\boldsymbol{o}_t^{(2)},\boldsymbol{s}_{t-1}^{(2)},\boldsymbol{a}_{t-1}))$$

$$\leq D_{\mathrm{KL}}(p_\theta(\boldsymbol{s}_t^{(1)}|\boldsymbol{o}_t^{(1)},\boldsymbol{s}_{t-1}^{(1)},\boldsymbol{a}_{t-1})||p_\theta(\boldsymbol{s}_t^{(2)}|\boldsymbol{o}_t^{(2)},\boldsymbol{s}_{t-1}^{(2)},\boldsymbol{a}_{t-1})) \tag{13}$$

Note that equality holds if the two distributions coincide. Analogously $I(S_t^{(2)};O_t^{(2)}|S_{t-1}^{(2)},A_{t-1},O_t^{(1)})$ is upper bounded by $D_{\mathrm{KL}}(p_\theta(\boldsymbol{s}_t^{(2)}|\boldsymbol{o}_t^{(2)},\boldsymbol{s}_{t-1}^{(2)},\boldsymbol{a}_{t-1})||p_\theta(\boldsymbol{s}_t^{(1)}|\boldsymbol{o}_t^{(1)},\boldsymbol{s}_{t-1}^{(1)},\boldsymbol{a}_{t-1}))$.

Assume $S_{t-1}$ is a sufficient representation of $O_{t-1}$. Then, $S_{t-1}^{(1)}$ provides task-relevant information no more than the sufficient representation $S_{t-1}$. $I(S_t^{(1)};O_t^{(2)}|S_{t-1}^{(1)},A_{t-1})$ can be thus re-expressed as:

$$I(S_t^{(1)};O_t^{(2)}|S_{t-1}^{(1)},A_{t-1})$$

$$\geq I(S_t^{(1)};O_t^{(2)}|S_{t-1},A_{t-1})$$

$$\overset{(\mathrm{P.2})}{=}I(S_t^{(1)};S_t^{(2)}O_t^{(2)}|S_{t-1},A_{t-1})-I(S_t^{(1)};S_t^{(2)}|O_t^{(2)},S_{t-1},A_{t-1})$$

$$\overset{*}{=}I(S_t^{(1)};S_t^{(2)}O_t^{(2)}|S_{t-1},A_{t-1})$$

$$=I(S_t^{(1)};S_t^{(2)}|S_{t-1},A_{t-1})+I(S_t^{(1)};O_t^{(2)}|S_t^{(2)},S_{t-1},A_{t-1})$$

$$\geq I(S_t^{(1)};S_t^{(2)}|S_{t-1},A_{t-1}) \tag{14}$$

Where $*$ follows from $S_t^{(2)}$ being the representation of $O_t^{(2)}$. The bound is tight whenever $S_t^{(2)}$ is sufficient from $S_t^{(1)}$ ($I(S_t^{(1)}; O_t^{(2)} | S_t^{(2)}, S_{t-1}, A_{t-1}){=}0$). This happens whenever $S_t^{(2)}$ contains all the information regarding $S_t^{(1)}$. Once again, we can have $I(S_t^{(2)}; O_t^{(1)} | S_{t-1}^{(2)}, A_{t-1}) \geq I(S_t^{(1)}; S_t^{(2)} | S_{t-1}, A_{t-1})$. Therefore, the averaged loss functions can be upper-bounded by

$$
\begin{aligned}
\mathcal{L}_{\frac{1+2}{2}} \leq & -\frac{\lambda_1 + \lambda_2}{2} I(S_t^{(1)}; S_t^{(2)} | S_{t-1}, A_{t-1}) \\
& + D_{\text{SKL}}(p_\theta(\boldsymbol{s}_t^{(1)} | \boldsymbol{o}_t^{(1)}, \boldsymbol{s}_{t-1}^{(1)}, \boldsymbol{a}_{t-1}) || p_\theta(\boldsymbol{s}_t^{(2)} | \boldsymbol{o}_t^{(2)}, \boldsymbol{s}_{t-1}^{(2)}, \boldsymbol{a}_{t-1}))
\end{aligned}
\tag{15}
$$

Lastly, by re-parametrizing the objective, we obtain:

$$
\begin{aligned}
\mathcal{L}(\theta; \beta) = & - I_\theta(S_t^{(1)}; S_t^{(2)} | S_{t-1}, A_{t-1}) \\
& + \beta D_{\text{SKL}}(p_\theta(\boldsymbol{s}_t^{(1)} | \boldsymbol{o}_t^{(1)}, \boldsymbol{s}_{t-1}^{(1)}, \boldsymbol{a}_{t-1}) || p_\theta(\boldsymbol{s}_t^{(2)} | \boldsymbol{o}_t^{(2)}, \boldsymbol{s}_{t-1}^{(2)}, \boldsymbol{a}_{t-1}))
\end{aligned}
\tag{16}
$$

In Algorithm 1, we use $\boldsymbol{s}_t^{(1)} \sim p_\theta(\boldsymbol{s}_t^{(1)} | \boldsymbol{o}_t^{(1)}, \boldsymbol{s}_{t-1}^{(1)}, \boldsymbol{a}_{t-1})$ and $\boldsymbol{s}_t^{(2)} \sim p_\theta(\boldsymbol{s}_t^{(2)} | \boldsymbol{o}_t^{(2)}, \boldsymbol{s}_{t-1}^{(2)}, \boldsymbol{a}_{t-1})$ to obtain representations for multi-view observations. We argue that the substitution does not affect the effectiveness of the averaged objective. With the multi-view assumption, we have that representations $\boldsymbol{s}_{t-1}^{(1)}$ and $\boldsymbol{s}_{t-1}^{(2)}$ do not share any task-irrelevant information. So, the representations at timestep $t$ conditioned on them do not share any task-irrelevant information. Maximizing the mutual information between $\boldsymbol{s}_t^{(1)}$ and $\boldsymbol{s}_t^{(2)}$ (first term in Eq. (16)) will encourage the representations to share maximal task-relevant information. Similar argument also works for the second term in Eq. (16). Since $\boldsymbol{s}_{t-1}^{(1)}$ and $\boldsymbol{s}_{t-1}^{(2)}$ do not share any task-irrelevant information, any task-irrelevant information introduced from the conditional probability will be also identified as task-irrelevant information by KL divergence, which will be reduced through minimizing the DRIBO loss.

## C   ADDITIONAL RESULTS

### C.1   ADDITIONAL DMC RESULTS

For clean setting, the pixel observations have simple backgrounds as shown in Figure 2 (left column). Figure 6 shows that RAD, SLAC, CURL and PI-SAC generally perform the best, whereas DRIBO consistently outperforms DBC and matches SOTA.

For natural video setting, Figure 7 shows that DRIBO performs substantially better than RAD, SLAC and CURL which do not discard task-irrelevant information explicitly. Compared to PI-SAC and DBC, recent state-of-art methods that aim at learning representations that are invariant to task-irrelevant information, DRIBO outperforms them consistently.

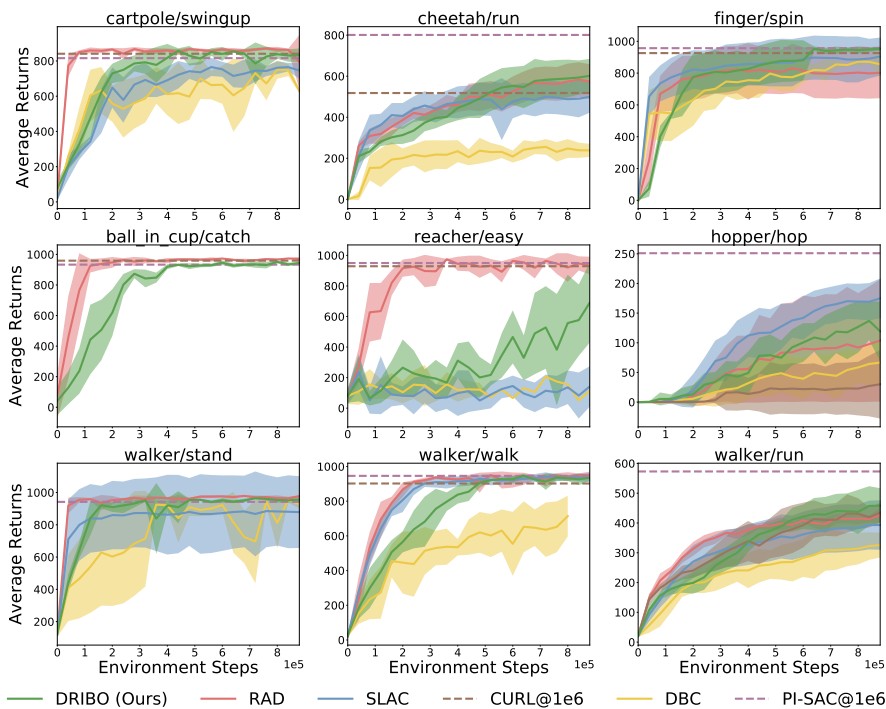

Figure 6: Average returns on DMC tasks over 5 seeds with mean and one standard error shaded in the clean setting.

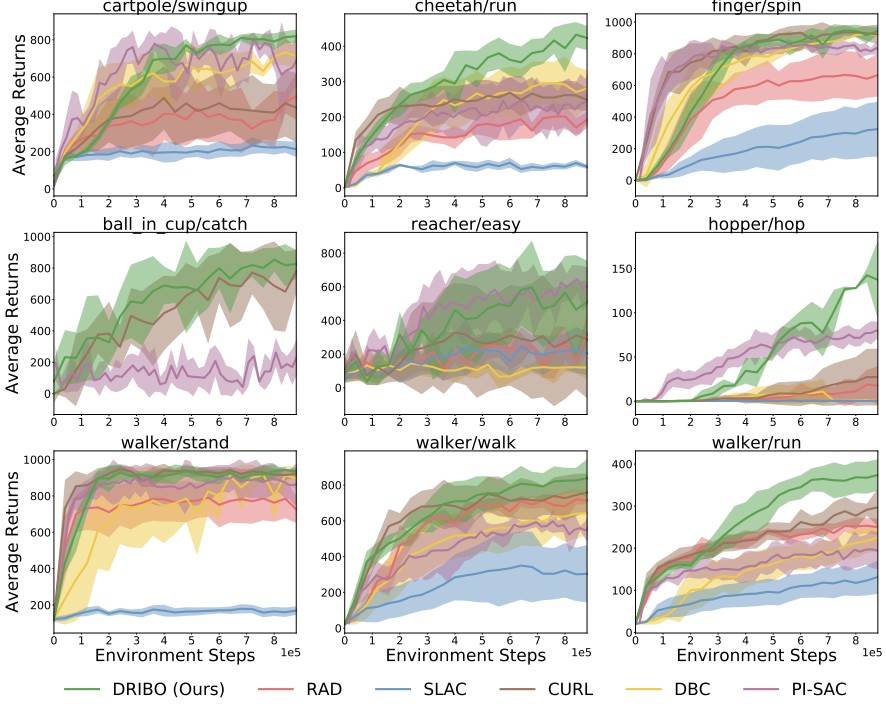

Figure 7: Average returns on DMC tasks over 5 seeds with mean and one standard error shaded in the natural video setting.

## C.2 DRIBO Loss vs. InfoMax

### C.2.1 Natural Video Setting

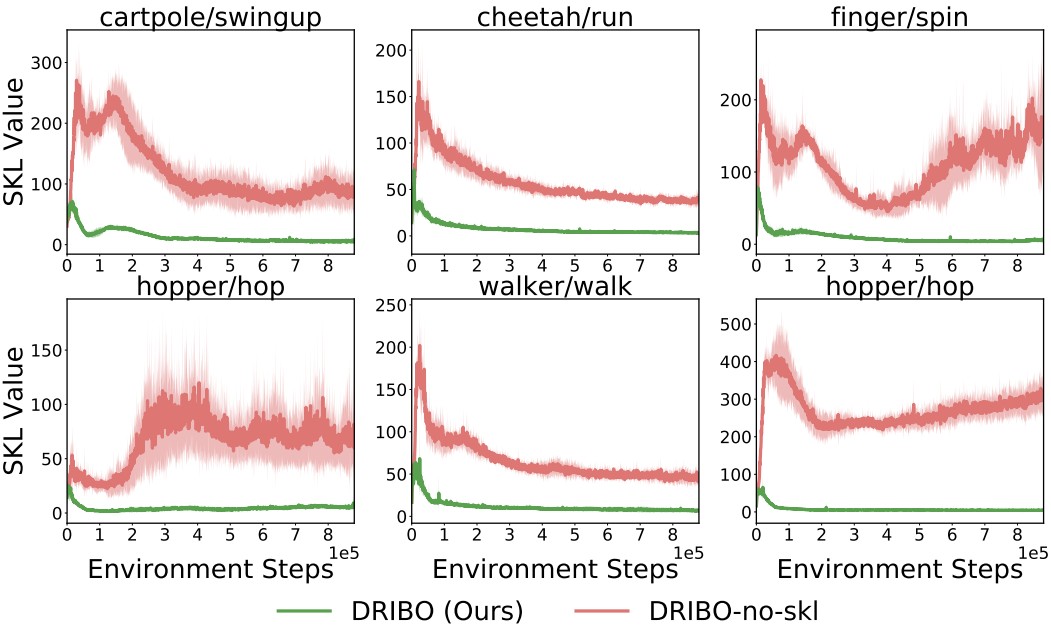

Figure 8: Average SKL values during training in DMC environments with natural videos as background.

To quantify the amount of task-irrelevant information retained in the representations, we compare the SKL term values between DRIBO and DRIBO-no-skl during training in Figure 8. As described in Section 5, DRIBO-no-skl is trained with an InfoMax-based objective without a SKL term. The gap between the SKL values explains the performance gap between the two approaches (as shown in Figure 5). The models trained with DRIBO take advantage of the information bottleneck to map observations from different views close to each other in the latent space. Figure 8 shows that minimizing the DRIBO loss consistently reduces the KL divergence between representations from different views. On the other hand, the models trained with DRIBO-no-skl fail to discard the task-irrelevant information contained in observations from different views even though the RSSM model helps to learn predictive representations. For DRIBO-no-skl, the KL divergence between representations from different views is consistently larger than the one learned by DRIBO.

### C.2.2 Clean Setting

We provide additional results on comparing DRIBO with DRIBO-no-skl under the clean setting Figure 9 and Figure 10. It can be observed that DRIBO and DRIBO-no-skl perform similarly in terms of average returns. Recall our earlier plot on comparing DRIBO with DRIBO-no-skl under the natural video setting Figure 5 which shows DRIBO substantially outperforms DRIBO-no-skl (in other words, the performance of DRIBO-no-skl drops when the setting is changed from clean to natural video). Similar results can be seen in Figure 6 and Figure 7, where the performance of approaches like RAD, SLAC, CURL and PI-SAC significantly degrades when the backgrounds of the environments are changed to different natural videos during training and testing.

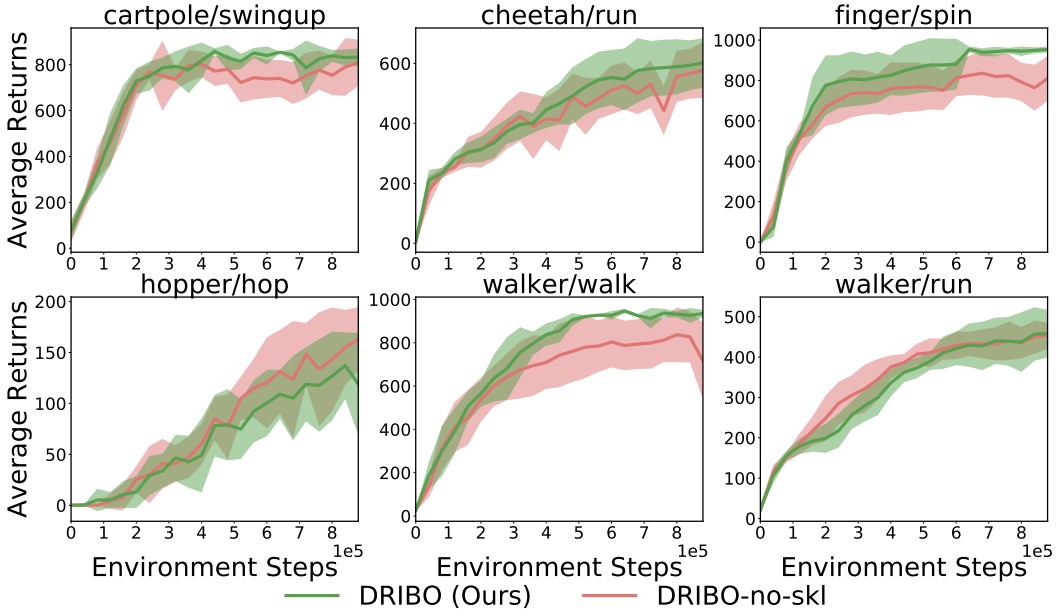

Figure 9: Average returns of DRIBO and DRIBO-no-skl on DMC tasks over 5 seeds with mean and one standard error shaded in the clean setting.

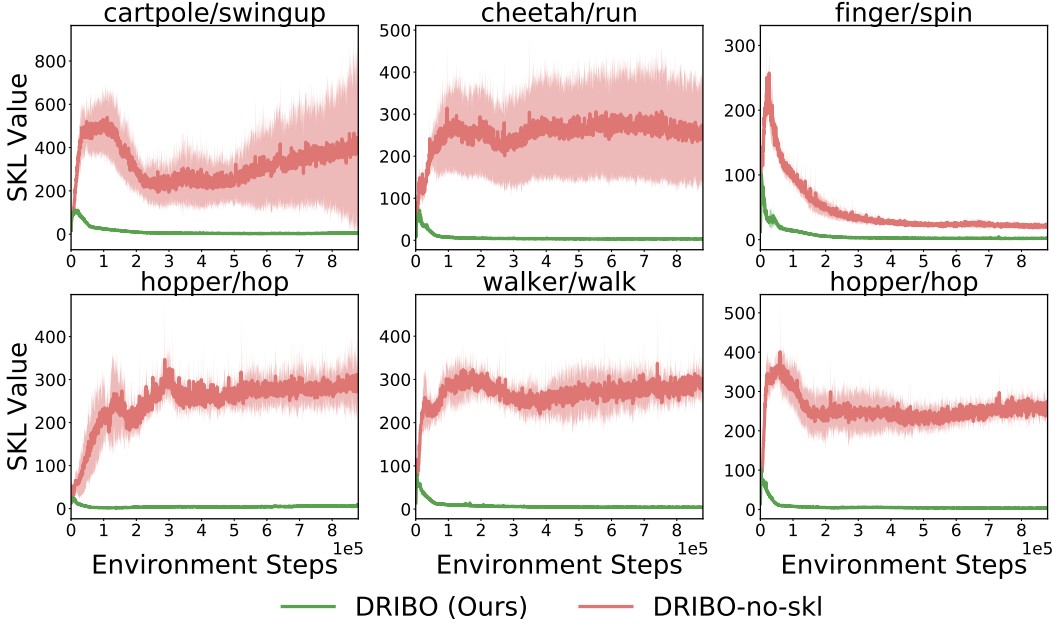

Figure 10: Average SKL values during training in DMC environments in the clean setting.

### C.3 TRAINING AND TESTING PERFORMANCE IN DMC UNDER THE NATURAL VIDEO SETTING

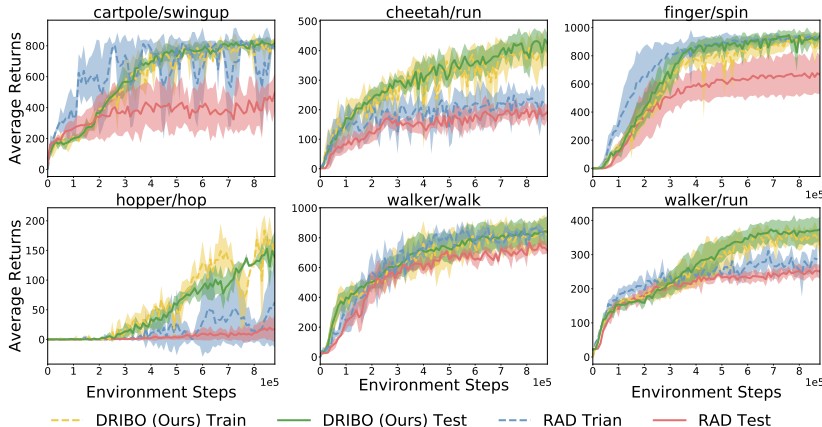

Figure 11: Training and testing performance of RAD and DRIBO.

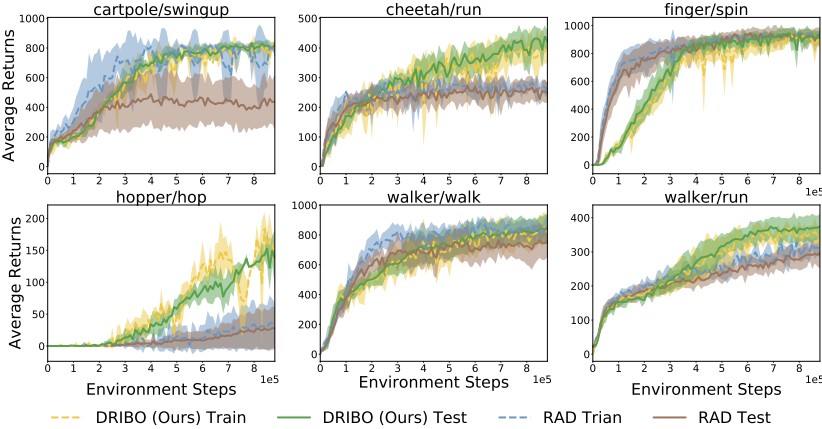

Figure 12: Training and testing performance of CURL and DRIBO.

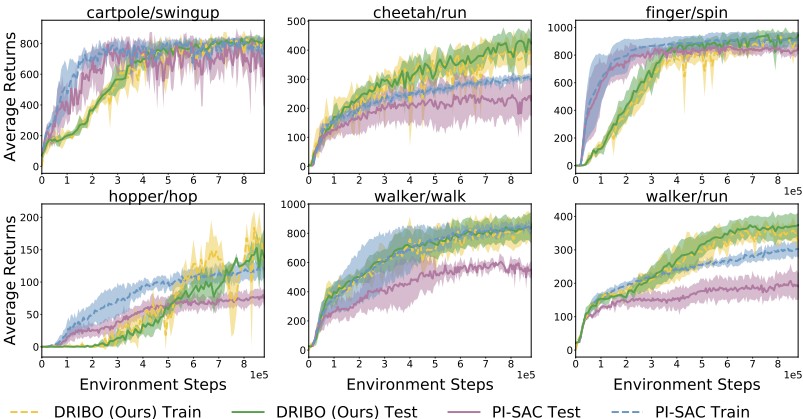

Figure 13: Training and testing performance of PI-SAC and DRIBO.

We further compare DRIBO with RAD, CURL and PI-SAC on the DMC environments under the natural video setting. We observe that RAD, CURL and PI-SAC could achieve high scores during training but failed to achieve the same high scores (with a substantial gap) during testing.

## C.4 ADDITIONAL VISUALIZATION

Figure 14 are the snapshots of training environments of DMC under the natural video setting. The background videos are randomly sampled from the class of "arranging flower" which are drastically different from backgrounds used during testing (Figure 15).

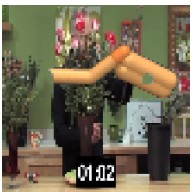 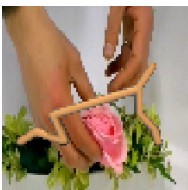 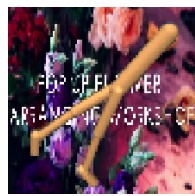 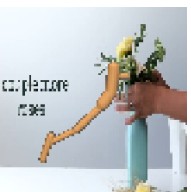 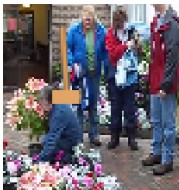

Figure 14: **Training environments of DMC under the natural video setting:** The background videos are sampled from arranging flower class in Kinetics dataset.

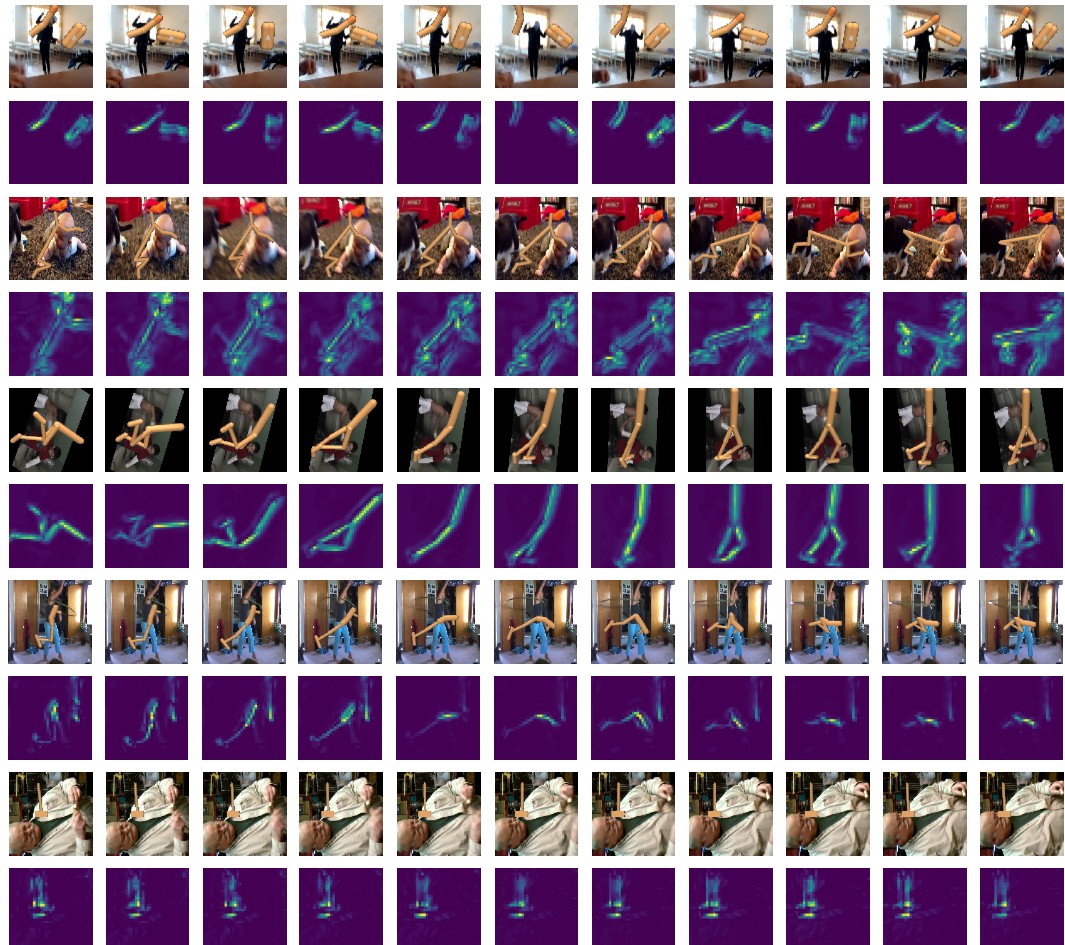

Figure 15: **Test environments of DMC under the natural video setting:** We show the sequential observations with natural videos as background in DMC and their corresponding spatial attention maps of the DRIBO trained encoder.

In Figure 15, we show sequential observations in test environments of DMC under the natural video setting. The backgrounds videos are randomly sampled from the test data of Kinetic dataset which contain various classes of videos. Note that a single run of the DMC task may have *multiple videos playing sequentially* in the background.

Figure 15 also visualizes the corresponding spatial attention maps of DRIBO trained encoders. For different tasks, DRIBO encoders' activations concentrate on entire edges of the body, providing a more complete and robust representation of the visual observations.

In addition to Figure 2, we also show spatial attention maps for each convolutional layers in Figure 16. We can observe that DRIBO trained encoders filter the complex distractors in the backgrounds gradually layer by layer.

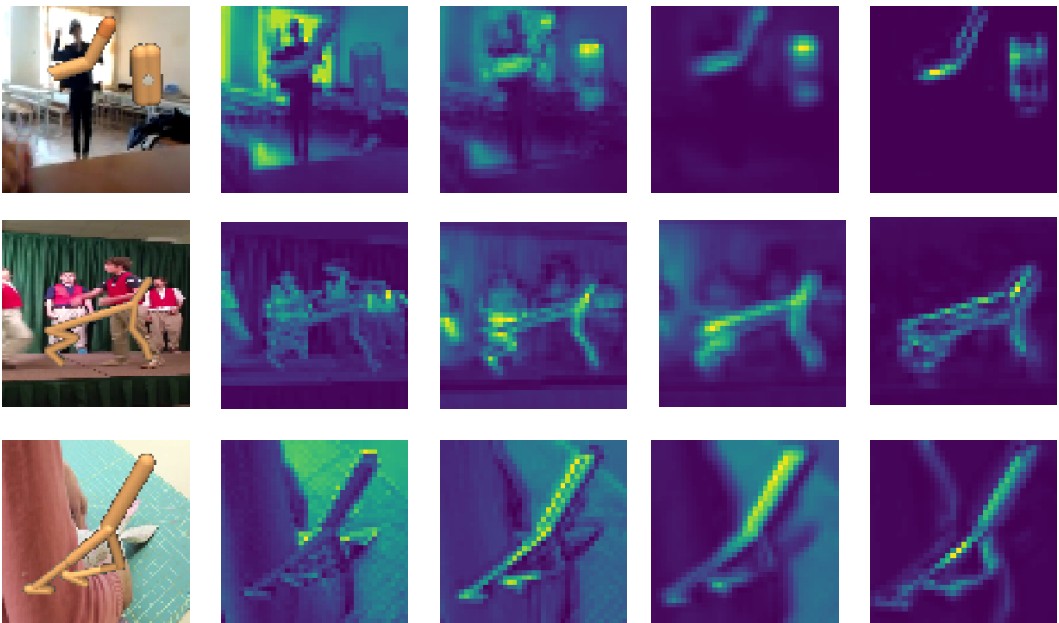

Figure 16: Spatial attention maps for each convolutional layers of the DRIBO trained encoders. The observations are the same as ones in Figure 2 from the snapshots during testing.

To further compare DRIBO with frame-stacked CURL, we visualize the learned representations using t-SNE plots in Figure 17. We see that even when the backgrounds are drastically different, our encoder learns to map observations with similar robot configurations near each other, whereas CURL's encoder maps similar robot configurations far away from each other. *This shows that CURL does not discard as many irrelevant features in the background as DRIBO does despite leveraging data augmentations and backpropagating RL objectives to the encoder.*

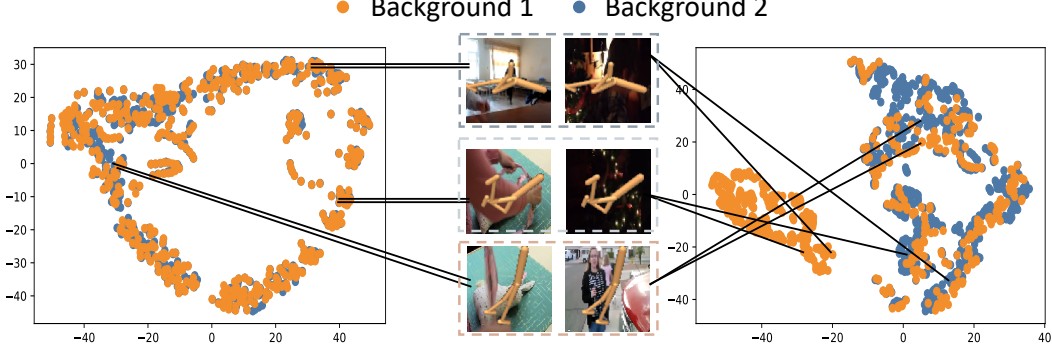

Figure 17: t-SNE of latent spaces learned with DRIBO (left) and CURL (right). They are the same as t-SNE in Figure 4. But we color-code the embedded points corresponding to their backgrounds. The observations are from the same trajectory but with different background natural videos (the same as in Figure Figure 4). Points from different backgrounds are close to each other in the embedding space learned by DRIBO, whereas no such structure is seen in the embedding space learned by CURL.

## D IMPLEMENTATION DETAILS

### D.1 RSSM

We split the representation $s_t$ into a stochastic part $z_t$ and a deterministic part $h_t$, where $s_t = (z_t, h_t)$. The generative and inference models of RSSM are defined as:

$$\text{Deterministic state transition: } h_t = f(h_{t-1}, z_{t-1}, a_{t-1})$$
$$\text{Stochastic state transition: } z_t = p(z_t|h_t)$$
$$\text{Observation model: } o_t = p(o_t|h_t, z_t)$$

where $f(h_{t-1}, z_{t-1}, a_{t-1})$ is implemented as a recurrent neural network (RNN) that carries the dependency on the stochastic and deterministic parts at the previous timestep. Then, we obtain the representation with the encoder $p_\theta((s_{1:T}|o_{1:T}, a_{1:T}) = \prod_t p_\theta(s_t|o_t, h_t)$, where $h_t$ retains information from $s_{t-1} = (z_{t-1}, h_{t-1})$ and $a_{t-1}$.

### D.2 DRIBO + SAC

We first show how we train SAC agent given the representations of DRIBO. Let $\phi(o) = s \sim p_\theta(s|o, s', a')$ denote the encoder, where $s'$ and $a'$ as the representation and action at last timestep.

---

**Algorithm 2** SAC + DRIBO Encoder

1: **Input** RL batch $\mathcal{B}_{\text{RL}} = \{(\phi(o_i), a_i, r_i, \phi(o'_i))\}_{i=1}^{(T-1)*N}$ with $(T-1)*N$ pairs of representation, action, reward and next representation.
2: Get value: $V = \min_{i=1,2} \hat{Q}_i(\hat{\phi}(o')) - \alpha \log \pi(a|\hat{\phi}(o'))$
3: Train critics: $J(Q_i, \phi) = (Q_i(\phi(o)) - r - \gamma V)^2$
4: Train actor: $J(\pi) = \alpha \log \pi(a|\phi(o)) - \min_{i=1,2} Q_i(\phi(o))$
5: Train alpha: $J(\alpha) = -\alpha \log \pi(a|\phi(o)) - \alpha \mathcal{H}(a|\phi(o))$
6: Update target critics: $\hat{Q}_i = \tau_Q Q_i + (1 - \tau_Q)\hat{Q}_i$
7: Update target encoder: $\hat{\phi} \leftarrow \tau_\phi \phi + (1 - \tau_\phi)\phi$

---

Then we incorporate the above SAC algorithm into minimizing DRIBO loss in Algorithm 3

---

**Algorithm 3** DRIBO + SAC

1: **Input**: Replay buffer $\mathcal{D}$ storing sequential observations and actions with length $T$. The batch size is $N$. The number of total training step is K. The number of total episodes is E.
2: **for** $e = 1, \ldots, E$ **do**
3:     Sample sequential observations and actions from the environment and append new samples to $\mathcal{D}$.
4:     **for** each step $k = 1, \ldots, K$ **do**
5:         Sample a sequential batch $\mathcal{B} \sim \mathcal{D}$.
6:         Compute the representations batch $\mathcal{B}_{\text{RL}}$ which has the shape $(T, N)$ using the encoder $p_\theta(s_{1:T}|o_{1:T}, a_{1:T})$
7:         Train SAC agent: $\mathbb{E}_{\mathcal{B}_{\text{RL}}}[J(\pi, Q, \phi)]$         ▷ Algorithm 2
8:         Update $\theta$ and $\psi$ to minimize $\mathcal{L}_{DRIBO}$ using $\mathcal{B}$         ▷ Algorithm 1.
9:     **end for**
10: **end for**

---

### D.3 DRIBO + PPO

The main difference between SAC and PPO is that PPO is an on-policy RL algorithm while SAC is an off-policy RL algorithm. With the update of the encoder, representations may not be consistent within each training step which breaks the on-policy sampling assumption. To address this issue, instead of obtaining $s_t$ propagating from the initial observation of the observation sequence, we store the representations as $s_t^{\text{old}}$ while sampling from the on-policy batch. Then, we use $\varphi(o) = s \sim$

$p_\theta(\boldsymbol{s}|\boldsymbol{o}, \boldsymbol{s}^{\text{old}}, \boldsymbol{a}')$ to denote the representation from the encoder. Here, $\boldsymbol{s}^{\text{old}}$ and $\boldsymbol{a}'$ are the representation and action at the previous timestep. By treating the encoding process as a part of the policy and value function, the on-policy requirement is satisfied since the new action/value at timestep $t$ depends only on $(\boldsymbol{o}_t, \boldsymbol{s}_{t-1}^{old}, \boldsymbol{a}_{t-1})$.

---

**Algorithm 4** DRIBO + PPO

---

1: **Input**: Replay buffer $\mathcal{D}$ and on-policy replay buffer $\mathcal{D}_{\text{PPO}}$ storing sequential observations and actions with length $T$. The batch size is $N$. The minibatch size for PPO is M. The number of total episodes is E.
2: **for** $e = 1, \ldots, E$ **do**
3:     Sample sequential observations and actions from the environment $\{(\boldsymbol{o}_{1:T}, \boldsymbol{a}_{1:T}, r_{1:T}, \boldsymbol{s}_{1:T}^{\text{old}}\}_{i=1}^N$.
4:     Append new samples to $\mathcal{D}$ and update the on-policy replay buffer $\mathcal{D}_{\text{PPO}}$.
5:     **for** $j = 1, \ldots, M$ **do**
6:         $\{(\varphi(\boldsymbol{o}_i), a_i, r_i)\}_{i=1}^{\lfloor \frac{T*N}{M} \rfloor} \sim \mathcal{D}_{\text{PPO}}$
7:         Optimize PPO policy, value function and encoder using each sample $(\varphi(\boldsymbol{o}_i), a_i, r_i)$ in the batch.
8:         Sample a sequential batch $\mathcal{B} \sim \mathcal{D}$.
9:         Update $\theta$ and $\psi$ to minimize $\mathcal{L}_{DRIBO}$ using $\mathcal{B}$       ▷ Algorithm 1.
10:     **end for**
11: **end for**

---

## D.4 DMC

We use an almost identical encoder architecture as the encoder in the RSSM paper (Hafner et al., 2019), with two minor differences. Firstly, we deploy the encoder architecture in Tassa et al. (2018) as the observation embedder, with two more convolutional layers to the CNN trunk. Secondly, we add layer normalization to process the output of CNN, deterministic output and stochastic output. Deterministic part of the representation is a 200-dimensional vector. Stochastic part of the representation is a 30-dimensional diagonal Gaussian with predicted mean and standard deviation. Thus, the representation is a 230-dimensional vector. We implement Q-network and policy in SAC as MLPs with two fully connected layers of size 1024 with ReLU activations. We estimate the mutual information using a bi-linear inner product as the similarity measure (Oord et al., 2018).

**Pixel Preprocessing.** We construct an observational input as a single frame, where each frame is a RGB rendering of size $100 \times 100$ from the 0th camera. We then divide each pixel by 255 to scale it down to $[0, 1)$ range. For methods compared in our experiments, an observation inputs contains an 3-stack of consecutive frames.

**Augmentations of Visual Observations.** For RAD, we use *random crop+ grayscale* to generate augmented data. For our approach DRIBO and CURL, we use *random crop* to generate multi-view observations. We apply the implementation of RAD to do the augmentation. For *random crop*, it extracts a random patch from the original observation. In DMC, we render $100 \times 100$ pixel observations and crop randomly to $84 \times 84$ pixels. For *random grayscale*, it converts RGB images to grayscale with a probability $p = 0.3$.

**Hyperparameters.** To facilitate the optimization, the hyperparameter $\beta$ in the DRIBO loss Algorithm 1 is slowly increased during training. $\beta$ value starts from a small value $1e - 4$ and increases to $1e - 3$ with an exponential scheduler. The same procedure is also used in the MIB paper (Federici et al., 2020). We show other hyperparameters for DMC experiments in Table 2.

**KL Balancing.** During training, we also incorporate *KL balancing* from a variant method described in DreamerV2 (Hafner et al., 2021) to train RSSM. *KL balancing* encourages learning an accurate prior over increasing posterior entropy, so that the prior better approximates the aggregate posterior. This help us to avoid regularizing the representations toward a poorly trained prior. *KL balancing* is orthogonal to our MIB objective (DRIBO Loss). Note that DreamerV2 deploy *KL balancing* based on a reconstruction loss. In our case, our results show that *KL balancing* can improve training of RSSM with a contrastive-learning-based or mutual-information-maximization objective. We implement

this technique as shown in Algorithm 5. $\beta$ shares the same value as the coefficient for SKL term in DRIBO Loss.

---

**Algorithm 5** DRIBO Loss + KL Balancing

---

1: Compute kl balancing term:

$$\text{kl\_balancing} = 0.8 \cdot \text{compute\_kl(stop\_grad(approx\_posterior)} + \text{prior)}$$
$$+ 0.2 \cdot \text{compute\_kl(approx\_posterior} + \text{stop\_grad(prior))}$$

**return** DRIBO Loss + $\beta \cdot$ kl_balancing

---

Table 2: Hyperparameters used for DMC experiments.

| Hyperparameters | Value |
|---|---|
| Observation size | $(100 \times 100)$ |
| Cropped size | $(84 \times 84)$ |
| Replay buffer size | 1000000 |
| Initial steps | 1000 |
| Stacked frames | No |
| Action repeat | 2 finger, spin; walker, walk; |
| | 8 cartpole swingup |
| | 4 otherwise |
| Evaluation episodes | 8 |
| Optimizer | Adam |
| Learning rate | encoder learning rate: 1e-5; |
| | policy/Q network learning rate: 1e-5; |
| | $\alpha$ learning rate: 1e-4. |
| Batch size | $8 \times 32$, where $T = 32$ |
| Target update $\tau$ | Q network: 0.01 |
| | encoder: 0.05 |
| Target update freq | 2 |
| Discount $\gamma$ | .99 |
| Initial temperature | 0.1 |
| Total timesteps | 88e4 |
| $\beta$ scheduler start episode | 10 |
| $\beta$ scheduler end episode | 60 |

### D.5 PROCGEN

For Procgen suite, the implementation of DRIBO is almost the same as DMC experiments. Better design choice could be found by validation. We use the same as the encoder architecture used in DMC experiments, except for the observation embedder, which we use the network from IMPALA paper to take the visual observations. In addition, since the actions in Procgen suite are discrete, we use an action embedder to embed discrete actions into continuous space. The action embedder is implemented as a simple one hidden layer resblock with 64 neurons. It maps a one-hot action vector to a 4-dimensional vector. The policy and value function share one hidden layer with 1024 neurons. The policy uses another fully connected layer to generate a categorical distribution to select the discrete action. The value function uses another fully connected layer to generate the value for an input representation. All activation functions are ReLU activations.

**Augmentation of Visual Observations.** We select augmentation types based on the best reported augmentation types for each environment. DrAC (Raileanu et al., 2020) reported best augmentation types for RAD and DrAC in Table 4 and 5 of the DrAC paper. We list the augmentation types used in DRIBO in Table 3 and 4. We use the same settings for each augmentation type as DrAC. Note that we only performed limited experiments to select the augmentations reported in the tables due to time constraints. So, the tables do not show the best augmentation types in each environment for DRIBO.

Table 3: Augmentation type used for each game.

| Env | BigFish | StarPilot | FruitBot | BossFight | Ninja | Plunder | CaveFlyer | CoinRun |
|---|---|---|---|---|---|---|---|---|
| Augmentation | crop | cutout | cutout | cutout | random-conv | crop | random-conv | random-conv |

Table 4: Augmentation type used for each game.

| Env | Jumper | Chaser | Climber | DodgeBall | Heist | Leaper | Maze | Miner |
|---|---|---|---|---|---|---|---|---|
| Augmentation | random-conv | crop | random-conv | cutout | crop | crop | crop | flip |

**Hyperparameters.** We use the same $\beta$ scheduler as the DMC experiments. The starting $\beta$ value is $1e-8$ and the final $\beta$ value is $1e-3$. We show other hyperparameters for Procgen environments in Table 5.

**Discussion.** Here, we extend the discussion on why our method underperforms on some environments, whose screenshots are shown in Figure 18.

Our approach, DRIBO, only consider global MI (Hjelm et al., 2019) in the current implementation. As a result, local structures can be easily ignored in the representation. More specifically, representations containing the same static local features within a single execution but at different timesteps are treated as negative examples in the mutual information maximization. Then, the information of these local features shared between representations is not maximized. Negative pairs of representations sharing this type of local features are globally negative pairs but locally positive pairs.

For Plunder, the goal is to destroy moving enemy pirate ships by firing cannonballs. The enemy ships can be identified with the color of the target in the bottom left corner. The background of the game maintains the same within a single execution. Wooden obstacles capable of blocking the player's cannonballs. In a single execution, critical features like the target label and wooden obstacles remain unchanged. Failing to capture these local features results in poor performance of an agent.

For Chaser, the goal is to collect all green orbs as well as stars. A collision with an enemy that is not vulnerable (red) results in the death of the player. The background remains the same across different executions. Walls in the environment are dense and remain static during a single execution. Failing to capture walls' positions in representations hinders the agent from navigating to avoid enemies and collect orbs/stars. In Maze and Leaper, walls also remain static, but the backgrounds are different across different executions. This difference reduces the influence introduced by globally negative pairs but locally positive pairs. By contrast, walls in DodgeBall remain static but more critical since hitting at a wall ends the game.

Table 5: Hyperparameters used for Procgen experiments.

| Hyperparameters | Value |
|---|---|
| Observation size | $(64 \times 64)$ |
| Replay buffer size | 1000000 |
| Num. of steps per rollout | 256 |
| Num. of epochs per rollout | 3 |
| Num. of minibatches per epoch | 8 |
| Stacked frames | No |
| Evaluation episodes | 10 |
| Optimizer | Adam |
| Learning rate | encoder learning rate: 1e-4; policy learning rate: 5e-4; $\alpha$ learning rate: 1e-4. |
| Batch size | $8 \times 256$, where $T = 256$ |
| Entropy bonus | 0.01 |
| PPO clip range | 0.2 |
| Discount $\gamma$ | .99 |
| GAE parameter $\lambda$ | 0.95 |
| Reward normalization | yes |
| Num. of workers | 1 |
| Num. of environments per worker | 64 |
| Total timesteps | 25M |
| $\beta$ scheduler start episode | 10 |
| $\beta$ scheduler end episode | 110 |

## D.6 COMPUTE RESOURCES AND LICENSE

**Compute Resources.** We used a desktop with a 12-core CPU and a single GTX 1080 Ti GPU for benchmarking. Each seed for DMC benchmarks takes 3 days to finish. For Procgen suite, it takes 2 days to finish experiments on each seed.

**License.** DMC benchmarks are simulated in MuJoCo 2.0. We perform the experiments in this paper under an Academic Lab License. We perform experiments in Procgen suite with its open source code under MIT License.

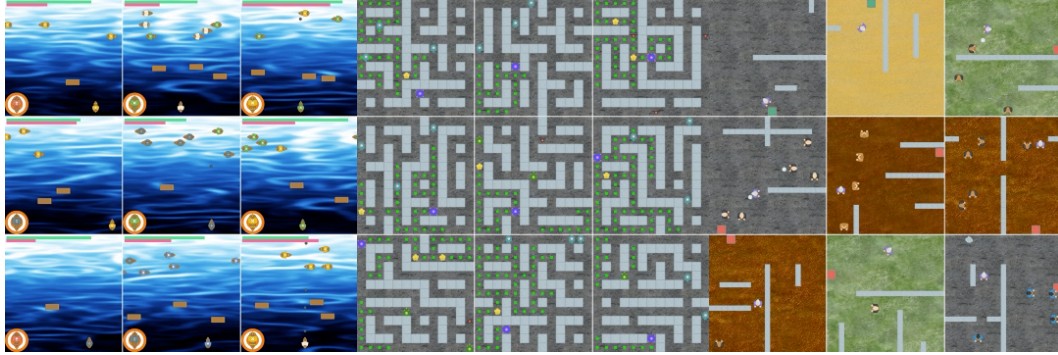

Figure 18: Screenshots of the three Procgen games where our approach DRIBO does not improve generalization performance compared to the other methods. From left to right, they are Plunder, Chaser and DodgeBall.

