# OpenReview forum: "DRIBO: Robust Deep Reinforcement Learning via Multi-View Information Bottleneck"
_ICLR.cc/2022/Conference — ICLR 2022 Submitted_

### Official Review · Reviewer_uNzG · 2021-10-30

**Correctness:** 4
**Technical Novelty And Significance:** 3
**Empirical Novelty And Significance:** 3
**Recommendation:** 6
**Confidence:** 4

**Main Review:**

1. Strengths and Weaknesses:

\+ The perspective of using mutual information to increase the robustness is interesting. The authors provide clear analyses of how to use mutual information to learn task-irrelevant information based on multi-view settings.

\+ The proposed method is a plug-in penalty that can be integrated into many modern RL algorithms.

\- The representation encoder and the estimator of mutual information (using InfoNCE) are borrowed from existing works, which may limit the contribution of this work.

\- The proposed methods show advantages in DMC and ProcGen environments. However, the settings in both environments are very similar: changing the background image in the same task. I am not sure if there are any real-world tasks that can be better solved by the proposed method. It would be great if the author can conduct experiments on other realistic tasks.


2. General questions

(1)	How is the data augmentation conducted to create the multi-view setting? For example, in Figure 1, is the entire image rotated or only the background rotated? If the entire image is rotated, will the state change and the action not be consistent with the state?

(2)	How to create the multi-view setting for layouts in the ProcGen suite? In my understanding, the layout of environments is corresponding to the difficulty level. How to conduct image augmentation on such difficulty levels? Or, do the author only change the background image in ProcGen?

(3)    Based on (2), I am also wondering if this method can be extended to more general settings to increase the robustness?

**Summary Of The Paper:**

Motivated by the fact that RL agents are sensitive to unseen environments, this paper proposes a method to separate task-relevant and task-irrelevant information from observations based on unsupervised multi-view settings. They use data augmentation methods to create multi-view data for calculating contrastive Multi-View Information Bottleneck (MIB) and use it as the objective to increase the generalization of model representation.

**Summary Of The Review:**

In general, I think the proposed idea of this paper is interesting, and the theoretical analysis seems correct to me. However, I still have questions about the details of how to implement the multi-view settings (In my general question (1)). Also, the experiment part seems a little weak since the two environment suites are very similar.

I think this paper is marginally above the borderline. I may raise my score if the author provides reasonable answers to my question.

---

> ### Author Response · Authors · 2021-11-15
> **Author Response to Reviewer uNzG**
>
> Thank you for the comments and we hope our response below will address your concerns.
>
> 1) Clarification on our contribution related to InfoNCE and the RSSM encoder.
>
> We would like to emphasize that InfoNCE and the RSSM encoder are implementation-specific choices in our work. Our contributions do not lie on the usage of these two approaches.
>
> First, our paper introduces a new objective, DRIBO loss, that maximizes the mutual information between sequences of observations and representations. The loss also compresses away task-irrelevant information shared between them for improving robustness of the representation. The main contribution is to factorize the mutual information between sequential data (Theorem 1) and construct the DRIBO loss given the factorized mutual information to maximize task-relevant information and compress away task-irrelevant information (Eq.7). InfoNCE is used to estimate the mutual information and can be replaced by any mutual information estimation method.
>
> Second, our paper leverages the temporal structure of RL and considers mutual information between sequential data. The RSSM encoder is not the only component that encourages capturing the temporal structure of RL. DRIBO loss is formulated to estimate the mutual information between sequential observations and representations. The encoder $p_\theta(s_t|o_t, s_{t-1}, a_{t-1})$ is also designed to learn the temporal structure. RSSM is an architecture that we used to train $p_\theta(s_t|o_t, s_{t-1}, a_{t-1})$.
>
> 2) Real-world tasks.
>
> DMC with the natural video setting is designed to evaluate RL agents on real-world data [1]. The natural videos are injected into DMC as signals from the natural world. The Procgen suite is designed to evaluate the generalization capability of RL agents by varying the difficulties and backgrounds of the games during training and testing [2]. Both benchmarks have been widely used in the RL community for these objectives and we believe they are representative.
> On the other hand, going beyond simulation environments is definitely a subject of our future investigations.
>
> 3) Generation of multi-view observations.
>
> In Figure 1, the background is rotated in the source natural video. We did not apply rotation as an augmentation to the background. In our paper, we mainly use *random crop* to augment an observation to create two random subimages of an original image as multi-view observations. The multi-view observations crop out different task-irrelevant information.
>
> 4) Multi-view setting for Procgen.
>
> We use the same type of data augmentation across all difficulty levels of a game in the Procgen experiments.
> Similar to the DMC experiments, we apply each type of stochastic data augmentations to get two random sub-images of an original observation as multi-view observations. The details are also described in Table 3 and 4.
>
> 5) Extend to more general settings to increase the robustness
>
> We would like to seek further clarification on the kind of "more general settings" the reviewer is considering here.
> In general, a multi-view setting can be constructed by applying different stochastic data augmentations.
>
> [1] Amy Zhang, Yuxin Wu, and Joelle Pineau. Natural environment benchmarks for reinforcement
> learning. arXiv preprint arXiv:1811.06032, 2018a.
>
> [2] Karl Cobbe, Christopher Hesse, Jacob Hilton, and John Schulman. Leveraging procedural generation
> to benchmark reinforcement learning. In International Conference on Machine Learning. PMLR,
> 2020.

---

> > ### Comment · Reviewer_uNzG · 2021-11-29
> > **Reply to the authors**
> >
> > Thank the authors for addressing my questions. Most of my concerns have been resolved.
> >
> > For point 3, I suggest the author revise Figure 1 if they don't use rotation in their method.
> >
> > For point 5, by "more general settings", I mean settings that are beyond the background image changing situation, for example, other types of irrelevant information in the task.

---

> > > ### Author Response · Authors · 2021-11-29
> > > **Post-rebuttal Response to Reviewer uNzG**
> > >
> > > We would like to thank the reviewer for the suggestions and clarifications!
> > >
> > > 1) Figure 1
> > >
> > > We will revise Figure 1 to avoid confusion.
> > >
> > > 2) More general settings
> > >
> > > DRIBO can be applied to settings beyond background changes. As long as the multi-view observations share the same task-relevant information, we can use DRIBO to identify and compress away information not shared between the multi-view observations. For example, the multi-view observations can be obtained from cameras with different poses. We evaluated DRIBO on background changes because they are commonly considered visual distractions and are easy to set up for experiments.

---

### Official Review · Reviewer_m1E2 · 2021-10-31

**Correctness:** 3
**Technical Novelty And Significance:** 3
**Empirical Novelty And Significance:** 2
**Recommendation:** 6
**Confidence:** 4

**Main Review:**

Strengths:
* The paper shows a technically approach to compress task-irrelevant information, making the agent focus on task-relevant information.
* The paper shows good empirical improvements on DM Control with natural video background and procgen, compared to existing methods in system-wise comparisons.
* Comprehensive ablation studies on DM Control that identify the importance of history length and information compression.

Weaknesses:
* It’s not clear to me whether there are as many task-irrelevant observation changes in procgen as in DM Control with video background. The generalization requirement in procgen seems to be different. I would like to see if the authors can clarify how the procgen experiment align with the main story of this paper.
* Many of DRIBO’s design choices (using RSSM, long history length, etc) are different to existing methods. The system-wise comparison on procgen (table 1) doesn’t provide much insight on why DRIBO improves generalization on procgen.
* The authors claim that $I(S_t^{(1)};S_t^{(2)}|S_{t-1},A_{t-1})$ can be maximized by maximizing $I(S_t^{(1)};S_t^{(2)})$. I don't think this is mathematically correct without making certain assumptions.

Smaller presentation issues:
* In Sec.3, the author states that "task-irrelevant information does not contribute to the choice of actions". I think it should be "task-irrelevant information does not affected by choice of actions".
* The author parameterizes mutual information as like $I_\theta$. Mutual information is a true measurement and can't be parameterized. Only MI estimates can be parameterized.
* CEB is a generic information bottleneck. It's incorrect to say that it cannot be applied to sequential data.
* The appendix shows in the main paper file.

**Summary Of The Paper:**

This paper studies the problem of pixel-based control with reinforcement learning. The authors categorize observation changes into task-relevant and task-irrelevant changes. They define task-irrelevant changes as those do not have casual relations with actions, and introduce a conditional prior to compress task-irrelevant information, such that agent can focus on task-relevant information. The authors evaluated the proposed DRIBO method on pixel-based control tasks (DM Control) with natural video background and the procgen suite.

**Summary Of The Review:**

The approach is novel. The paper shows good empirical results on DM Control with natural video background and procgen. However, it's not clear to me why the procgen-type of generalization and the procgen experiment are relevant to the main story and I have some concerns about the mathematical correctness. I'm slightly leaning to acceptance at this point, but I hope the authors would be able to clarify.

---

> ### Author Response · Authors · 2021-11-15
> **Author Response to Reviewer m1E2**
>
> We thank the reviewer for the comments and we hope our response below will address the questions raised in the review.
>
> 1) Source of task-irrelevant information in Procgen
>
> We use data augmentation to generate multi-view observations. We use the same stochastic data augmentation implementation as in RAD, DrAC and UCB-DrAC to generate multi-view observations from one original observation. The stochasticity of the chosen data augmentations generates random subimages of an original image as multi-view observations. It ensures that the multi-view observations are different with each other and contain different amounts/kinds of task-irrelevant information from the original observation. For example, applying *random crop* crops out different task-irrelevant information in the original observation.
>
> 2) Ablations on DRIBO.
>
> In Figure 5, we show results of careful ablations on the two main contributions of our paper.
> We first present ablations on the learning objectives. Agents trained with DRIBO loss outperform the one trained with just the InfoMax objectives (which is equivalent to the DRIBO loss without the SKL term as stated in the paragraph above Section 5.3).
> We also show ablations on DRIBO agents trained using sequences of different lengths under the natural video setting. It can be observed that DRIBO performs significantly better when training with longer sequences as longer sequences carry more temporal information.
>
> Our settings for the Procgen experiments are almost identical to the settings in the DMC experiments. We believe the conclusions dranw from the ablated studies on the DMC benchmarks with results shown in Figure 5 would also apply to the Procgen experiments.
>
> 3) $I(S_t^{(1)}; S_t^{(2)})$
>
> We thank the reviewer for pointing this out. We will correct the sentence "The mutual information between the two representations $I_\theta(S_t^{(1)}; S_t^{(2)} | S_{t - 1}, A_{t - 1})$ can be maximized by using any sample-based differentiable mutual information lower bound $I_{\psi}(S_t^{(1)}, S_t^{(2)})$, where $\psi$ represents the learnable parameters." as "The mutual information between the two representations $I_\theta(S_t^{(1)}; S_t^{(2)} | S_{t - 1}, A_{t - 1})$ can be maximized by using any sample-based differentiable mutual information lower bound $I_{\psi}(s_t^{(1)}, s_t^{(2)})$, where $\psi$ represents the learnable parameters." $s_t^{(1)}$ and $s_t^{(2)}$ are samples generated by the encoder $p_\theta(s_t|o_t, s_{t-1}, a_{t-1})$ which encodes the conditional information into representations.

---

> > ### Comment · Reviewer_m1E2 · 2021-11-25
> > **Post-rebuttal comments**
> >
> > **ProcGen experiments**
> >
> > The authors' response didn't answer my question. It's obvious to see that task-irrelevant information in DMC with distractors comes from the distractors. It's not obvious for me to see what's "the source of task-irrelevant information" in ProcGen. The authors responded that randomly generated data augmentation can lead to different amount of task-irrelevant information that the observation contains, which I agree, but doesn't tell where the task-irrelevant information comes from. I think Reviewer LDpb also raised a similar concern -- the ProcGen analysis is quite thin.
> >
> > **$I(S_t^{(1)}, S_t^{(2)})$**
> >
> > Now I can see that it's because the authors define the following Markov chains: $S_{t-1}, A_{t-1} \rightarrow S_t^{(1)}$, $S_{t-1}, A_{t-1} \rightarrow S_t^{(2)}$. However, I would again like to point out that it doesn't seem common to write the parameterized MI lower bound as $I_\psi$ and you can't write $s_t^{(1)}, s_t^{(2)}$ without expectation in it either for a sampling-based lower bound. I believe that the authors would be able to fix this though.
> >
> > Overall, the authors' response didn't completely resolve my concern, so I decided to keep my initial score at 6.

---

> > > ### Author Response · Authors · 2021-11-25
> > > **Post-rebuttal Response to Reviewer m1E2**
> > >
> > > We would like to thank the reviewer again for the comments!
> > >
> > > 1) Source of the task-irrelevant information in the Procgen Suite.
> > >
> > > In all the environments of the Procgen suite, procedural generation controls not only the selection of game assets (e.g. icons of fruits in FruitBot) but also the backgrounds [2]. Some environments include a more diverse pool of backgrounds than the others, e.g., FruitBot, BossFight, Ninja, CaveFlyer, and Jumper, etc. DRIBO performs better than other approaches in these environments. Similar to the background distractors in DMC, these backgrounds do not contribute to the rewards and they are not affected by the agent's actions.
> > >
> > > 2) $I(S_t^{(1)}; S_t^{(2)})$
> > >
> > > We thank the reviewer for pointing this out. In the revision, we will add the lower bound estimation as follows.
> > >
> > > $$-E_{p_\theta(s_t^{(1)}, s_t^{(2)}| o_t^{(1)}, o_t^{(2)}, s_{t-1}^{(1)}, s_{t-1}^{(2)}, a_{t-1})}E_{S^{-}}[\log \frac{exp(\phi(s_t^{(1)}, s_t^{(2)}))}{\sum_{s' \in S^{-} \cup s_t^{(2)}} exp(\phi(s_t^{(1)}, s'))}]$$
> > >
> > > where $S^-$ denotes a set of negative samples and $\phi$ is a function that outputs a scalar-valued score.
> > >
> > > [2] Karl Cobbe, Christopher Hesse, Jacob Hilton, and John Schulman. Leveraging procedural generation
> > > to benchmark reinforcement learning. In International Conference on Machine Learning. PMLR,
> > > 2020.

---

### Official Review · Reviewer_kUtq · 2021-11-02

**Correctness:** 3
**Technical Novelty And Significance:** 2
**Empirical Novelty And Significance:** 2
**Recommendation:** 5
**Confidence:** 5

**Main Review:**

The paper is well motivated and the main problem is interesting in my opinion. The related work is well covered and most of the writing flows well. Experimenting over two benchmarks (ProcGen + DMC) is quite extensive relative to other similar papers.

I start to have issues in understanding around Definition 1. I get the overall concept, that we can deconstruct the overall mutual information in a task-relevant and a task-irrelevant terms. However, why should we maximize the sum of both (as noted in the 5th line below Eq. 2)? You say maximizing I(O, S) gets you a sufficient representation. That would mean maximizing the first term as well. But you also say that the first term needs to be minimized. This seems contradictory. I understand Theorem 1 and also get the derivation of the components of Eq. 5 and 6. However in Eq. 5, do we want to maximize both terms? One is the task irrelevant term (aka minimality) while the other is the task relevant term (aka sufficiency). They should have opposite signs right (as is in Equation 7)?

Finally, regardless of the derivation of the DRIBO loss, my main concern is the empirical results here. The scores reported for DMC are not accurate in my opinion. I have run RAD on the distractor suite and it gets much better performance than what is reported. Why is RAD performance this low? DRIBO is using augmentations from RAD to generate the multi-view observations, and so as a first baseline, it should be performing better than RAD. Note that RAD achieves the same score (roughly) as reported in the original paper even when run with distractors. Can the authors clarify this?

Moreover, the performance reported for the proposed method is not good enough. In particular, running SAC with a reward prediction head achieves similar or even better performance than of the proposed method. In light of these two observations, I am not convinced that DRIBO really is doing more than a simple baseline, and certainely not doing so much better than RAD.

For the ProcGen experiments, why don't you compare with DAAC [1] and IDAAC [1] since I believe they are the state of the art methods on ProcGen currently.

Why compare the t-sne with CURL? We already know that CURL does not do a good job and that RAD is much better and even simpler. Also, in Figure 3, CURL is outperforming RAD in some of the tasks, which I think is again almost certainly not the case (based on the experiments I have run with RAD).

References:

[1] Decoupling Value and Policy for Generalization in Reinforcement Learning

**Summary Of The Paper:**

This paper tackles the problem of generalization amidst visual distractors for control tasks. In particular, the distractors have no dependence on the optimal policy and thus clearly form a task-irrelevant component. The proposal is to use mutual information between two views as a proxy for how much task-relevant information is present in the constructed representation. This objective is adapted for RL to consider the long term sequential nature. Finally, the authors test this approach on ProcGen and DMC Suite with distractors, while also performing certain ablations.

**Summary Of The Review:**

The paper is motivated by an interesting problem to the community, proposes a fairly novel method for leraning better representations, but there remain certain gaps in my understanding currently. I hope the authors can clarify this during the rebuttal. In any case, the experimental results are not correct in my prior experience of running these exact methods. For this reason, I am advocating for a reject. I'm happy to revise my review if these issues can be addressed.


-------------------------- Post Rebuttal ----------------------------------

The new results provided by the authors convince me fairly that the method is somewhat useful. I am still skeptical about the RAD results and the overall scores reported for DMC tasks, since the absolute numbers are not state-of-the-art. This might be due to a different evaluation setting where the distractor videos are changing across training, and at test time a video is drawn from a different distribution. I am unsure how much this sort of an evaluation scheme affects RAD's results, which should produce much better numbers than reported. I am giving the benefit of doubt here to the authors and hoping that they have done a fair evaluation for the baseline RAD method.

---

> ### Author Response · Authors · 2021-11-15
> **Author Response to Reviewer kUtq**
>
> Thank you for the comments and we hope our response below will address your concerns.
>
> 1) Clarification on the derivation of the DRIBO loss.
>
> "Maximizing
> $I(O_{1:T}; S_{1:T})$ learns a sufficient representation." is a statement that says the learned representations are sufficient for control (as defined in Definition 1) even though maximizing the first term in Eq.2 may make the representations non-robust. The downside of solely maximizing the mutual information is that it does not compress away any task-irrelevant information from observations. The observations from Eq.2 leads to one of the main contributions of our work. While maximizing $I(O_{1:T}; S_{1:T})$, the first term in Eq.2 does not contribute to making the representations sufficient since it represents the task-irrelevant information. By leveraging the multi-view information bottleneck principle, we can construct a loss (i.e. the DRIBO loss) to maximize the mutual information between representations and observations and compress away task-irrelevant information (the first term in Eq.2) in the representations to make them robust.
>
> Eq.5 and Eq.6 are losses constructed with the multi-view information bottleneck. They maximize the amount of task-relevant information shared between representations and observations, and compress away task-irrelevant information.
> Note that the first and the second terms have opposite signs within the parentheses and there is a minus sign in the front of the left parenthesis.
>
> 2) DMC with natural video is not the same as the DMC distraction suite.
>
> In our paper, we use a setting similar to but still somewhat different from the setting in the Distracting Control Suite (mainly because the Distracting Control Suite had not been published when we started on these experiments). We describe our natural video setting in Section 5.1. We choose a specific class ("arranging flowers") of Kinetics dataset to replace the backgrounds during training. Snapshots of the training environments can be found in Figure 14. During testing, to test whether the trained DRIBO agent can generalize to environments with unseen backgrounds, we replace the backgrounds of DMC with videos randomly chosen from the whole test set in the Kinetics dataset. These videos cover a wide range of classes. Sample snapshots during testing can be found in Figure 15.
>
> We compare DRIBO with RAD, CURL, DBC and PI-SAC on the DMC environments under the natural video setting. We observe that RAD could achieve high scores during training but
> failed to achieve the same high scores (with a substantial gap) during testing [Figure](https://1drv.ms/b/s!AhlLtryzQXzwkkmQy9uqvu9FXX44?e=V37Eop).
> We observe a similar performance gap for CURL and PI-SAC.
> We will include a discussion of this and the additional figures (RAD, CURL and PI-SAC) in the Appendix of the revision.
>
> We appreciate the reviewer's comments and hope our clarification can address the concerns regarding the RAD results.
>
> 3) Comparison with DAAC and IDAAC.
>
> We thank the reviewer for pointing us to the DAAC and IDAAC work (the first versions of DRIBO and IDAAC were actually published on arXiv within days of each other). We provide an additional comparison in [this table](https://1drv.ms/b/s!AhlLtryzQXzwkkvxkb7IMrYaP55K?e=7lje0y). DRIBO outperforms DAAC and IDAAC in *9 of the 16 games*.
> We will include these results in the revision.
> We pick the current set of methods (RAD, DrAc and UCB-DrAC) as baselines because they all leverage data augmentations. Comparing with these baselines helps to isolate the effect of the DRIBO loss since DRIBO uses the same data augmentations as these baselines do.
> % to show that the DRIBO loss and the DRIBO representation method outperform these baselines.
>
> 4) t-SNE with CURL.
>
> We compare the t-SNE with CURL since CURL is a good baseline comparison for DRIBO.
> DRIBO differs from CURL from mainly two aspects and these two differences are the main contributions of DRIBO.
> The first is that the learning objectives are different (InfoMax vs. DRIBO loss). We leverage the multi-view mutual information bottleneck to design the DRIBO loss. Second, DRIBO is trained with sequential data whereas CURL is trained from single-step observations. In contrast, RAD is trained with an RL objective only and does not specify a latent space.
> We will clarify this further in the revision.

---

> > ### Comment · Reviewer_kUtq · 2021-11-27
> > **Thank you for the response!**
> >
> > It's interesting that there is this gap in training and test performance for RAD. My hunch is that you could probably get better RAD performance by only using crop augmentation. The absolute numbers you are reporting are still not state-of-the-art in my opinion. But I am fairly satisfied with the new results (esp. ProcGen) and am now fine with this paper getting accepted, simply because it does a good job overall.

---

> > > ### Author Response · Authors · 2021-11-28
> > > **Post-rebuttal Response to Reviewer kUtq**
> > >
> > > We would like to thank the reviewer again for your insights and suggestions!
> > >
> > > We re-ran RAD with crop on three environments of DMC under the natural video setting: Cheetah Run, Hopper hop and Walker Walk. The results can be found [here](https://1drv.ms/b/s!AhlLtryzQXzwkkzNWZVZOkzJ4dU_?e=hXYcNc). We compare the training and testing performance of RAD using crop with DRIBO. In our submission, we used RAD with crop-grayscale since it is reported as the best augmentation method in the original RAD paper. To compare the difference between RAD using crop-grayscale and RAD using crop, we plot the results for both versions of RAD in this figure. We also plot the results of CURL in the figure as a reference.
> > >
> > > We want to highlight that DRIBO still achieves the best performance among all the methods. RAD using crop achieves better performance and outperforms CURL in these three environments. However, RAD still has a larger performance gap between training and testing. We will include the results of RAD using crop in our revision.

---

### Official Review · Reviewer_LDpb · 2021-11-08

**Correctness:** 3
**Technical Novelty And Significance:** 3
**Empirical Novelty And Significance:** 3
**Recommendation:** 5
**Confidence:** 3

**Main Review:**

Strength:
- To me, the final experiment result on ProcGen is impressive.
- Supplementary material shows that the authors seem to have digged into the experiment data with some depth.

Weakness:
- The writing is not so clean. Needs polishing or even rewriting.
  - The paper introduction is quite vague and covers very general ideas, lacking specificity and advantage to the proposed approach;
  - In Eq (5) and (6), there abruptly comes the Lagrange multiplier. Many other places also show that the writing is very rough.
- The results on DeepMind Control Suite are somewhat marginal improvement.
- While the results on ProcGen are impressive, the analysis and interpretation into these results are somewhat thin. Because the method involves many differences in comparison to the baselines, it is unclear whether the proposed main idea, mutual information loss, is the source of the major contributions. To me, the breakdown of contributions of each design choice (esp. including fair comparison with baselines with the same type and level of data augmentation) is important.

**Summary Of The Paper:**

This paper aims to learn robust representation from the replay buffer for reinforcement learning. The key idea is to leverage the concept of mutual information and the InfoNCE tool to compute the mutual information as a regularizer (a.k.a. DRIBO loss). The authors conducted experiments on some standard benchmarks (DeepMind Visual Control Suite and ProcGen).

**Summary Of The Review:**

Although there might be some good core contribution, the paper in its current state is not quite ready for publication. See details in my Main Review.

---

> ### Author Response · Authors · 2021-11-15
> **Author Response to Reviewer LDpb**
>
> Thank you for the comments and we appreciate the feedback. We hope our response below will address your concerns.
>
> 1) Ablation study on the source of good performance
>
> In Figure 5, we show careful ablation analysis on the two main contributions of our paper.
> We first present ablations on the learning objectives. Agents trained with the DRIBO loss outperform the ones trained with just the InfoMax objective (which is equivalent to the DRIBO loss without the SKL term as stated in the paragraph above Section 5.3).
> We also show ablations on DRIBO agents trained using sequences of different lengths under the natural video setting. It can be observed that DRIBO performs significantly better when training with longer sequences since longer sequences carry more temporal information.
>
> Regarding the comment on comparison with baselines with the same type and level of data augmentation, we would like to point the reviewer to the second paragraph in Section 5.3 where we state "For RAD and DrAC, we use the best reported augmentation types [in Table 4 and Table 5 of the DrAC paper] for different environments. DRIBO selects the same augmentation types except for a few games." and Appendix D.5 where we detail all the augmentation types that we used for each game.
>
> DRIBO's settings for the Procgen experiments are almost identical to the settings in the DMC experiments. We believe the conclusion drawn from the ablated studies on the DMC benchmarks with results shown in Figure 5 would also apply to the Procgen experiments.

---

### Author Response · Authors · 2021-11-22
**List of revisions**

In the revised submission, we have made the following changes.

1) We have included a discussion of DAAC [1] and IDAAC [1] in our related work section. We have also included the comparison between DRIBO and these two methods in Table 1 for the ProcGen experiments.

2) We have added Appendix C.3 to show results of training and testing performance on DMC under the natural video setting.

3) We have fixed all the other minor issues pointed out by the reviewers.

[1] Decoupling Value and Policy for Generalization in Reinforcement Learning, Raileanu et al. ICML 2021

---

### Decision · Program_Chairs · 2022-01-20

**Decision:**

Reject

**Comment:**

The authors introduce a method that improves the representation learned by RL agents, making them more robust to visual distractions. In particular, their approach proposes to use mutual information between two views as a proxy for that objective. This is clearly a borderline paper that required many discussions among the reviewers and the authors. The reviewers mention that the approach is novel, addresses an important problem of robustness in RL and some of the experiments provided are impressive. On the other hand, the reviewers point out that the baselines seem to achieve lower results than previously reported, writing could be improved and some of the results don't show significant improvement over baselines.

Given that some of the results cause confusion around the evaluation protocol (it's still not 100% clear why the performance of baselines is lower than expected) and other doubts expressed by the reviewers, I encourage the authors to continue working on the paper and resubmit. I believe that with a little bit of extra work and clarifications this can be a very strong submission.